# In Good GRACEs: Principled Teacher Selection for Knowledge Distillation

**Abhishek Panigrahi**[†] **Bingbin Liu**[α] **Sadhika Malladi**[θ] **Sham Kakade**[α] **Surbhi Goel**[ς]
[†]Princeton Language and Intelligence   [α]Kempner Institute, Harvard
[θ]Microsoft Research, New York   [ς]University of Pennsylvania

## Abstract

Knowledge distillation is an efficient strategy to use data generated by large "teacher" language models to train smaller capable "student" models, but selecting the optimal teacher for a specific student-task combination requires expensive trial-and-error. We propose a lightweight score called GRACE to quantify how effective a teacher will be for post-training a student model. GRACE measures distributional properties of the student's gradients without access to a verifier, teacher logits, teacher internals, or test data. From an information-theoretic perspective, GRACE connects to leave-one-out stability of gradient-based algorithms, which controls the generalization performance of the distilled students. On GSM8K and MATH, GRACE correlates strongly (up to $86\%$ Spearman correlation) with the performance of the distilled LLaMA and OLMo students. In particular, training a student using the GRACE-selected teacher can improve the performance by up to $7.4\%$ over naively using the best-performing teacher. Further, GRACE can provide guidance on crucial design choices in distillation, including (1) the best temperature to use when generating from the teacher, (2) the best teacher to use given a size constraint, and (3) the best teacher to use within a specific model family. Altogether, our findings demonstrate that GRACE can efficiently and effectively identify a strongly compatible teacher for a given student and provide fine-grained guidance on how to perform distillation.

## 1 Introduction

Distillation is an efficient and effective method to produce capable small models from existing, powerful teacher models. In this work, we focus on the specific case of training autoregressive language models on text generated by a teacher model. It is difficult to select the right teacher for a given student and task: a counterintuitive fact is that *a stronger-performing model is not always a better teacher*, which has been observed in classic classification/regression settings (Mirzadeh et al., 2019; Harutyunyan et al., 2023; Panigrahi et al., 2025) and more recently in the context of language models (Zhang et al., 2023; 2025a; Peng et al., 2025; Razin et al., 2025). Given the large number of available models as potential teachers, the current approach of guess-and-check is costly, because it requires collecting generations from a capable teacher and subsequently training a student on those generations. Additionally, the specific hyperparameters used in both phases can dramatically affect the final performance of the student, underscoring the need for careful, repeated testing to select the right teacher. As such, the current work seeks to address the following question:

*Given a pool of candidates, can we efficiently identify the best teacher for a given student and task?*

We argue that identifying a good teacher requires jointly considering the teacher and the student. To this end, we study the teacher's influence on the student's optimization process by analyzing the student's gradients. We propose a score, **GRACE** (GRAdient Cross-validation Evaluation), which captures the distributional properties of the student's gradients on a small set of teacher-generated data, enabling efficient and effective identification of the most compatible teacher (Section 2.2). Inspired by prior data selection and distillation works, GRACE unifies data diversity and teacher-student

---

Correspondence to: `ap34@princeton.edu`, `bliu@g.harvard.edu`.
Code is available at https://github.com/abhishekpanigrahi1996/GRACE

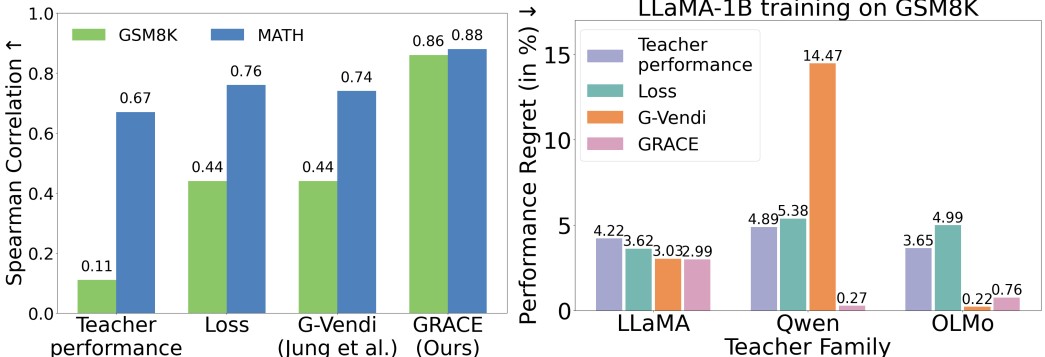

Figure 1: **GRACE correlates with student performance after distillation on math-related reasoning tasks.** We evaluate LLaMA-1B and LLaMA-3B students on GSM8K and MATH respectively with 15 teachers across LLaMA, Gemma, Qwen, OLMo, and Phi families. (Left) *GRACE shows the strongest correlation with student accuracy* among four scores—teacher performance, student's loss (before training) on the teacher generations, G-Vendi, and GRACE. (Right) *GRACE reliably selects near-optimal teachers within each teacher family*, measured by its small *teacher-student regret*, which is the absolute gap in final performance between the best overall student and that obtained from the teacher chosen by each score. Performance is measured by average accuracy over 16 generations per prompt.

alignment desiderata into a single score. Specifically, GRACE leverages a cross-validation structure, where it takes gradients from one split of data and measures their magnitudes re-weighted under the spectrum of the other split. The gradient spectrum reflects data diversity, while the gradient norm encodes teacher-student alignment. This cross-validation structure also allows us to draw a natural connection to conditional mutual information-based generalization bounds (Steinke & Zakynthinou, 2020; Rammal et al., 2022), providing insight into why GRACE works (Lemma 1). Importantly, GRACE is efficient to compute and does not rely on access to an external verifier, teacher logits, teacher representations, or test data.

We conduct thorough experiments to verify that the GRACE scores of teachers accurately reflect the student's performance when trained on those teachers. We focus on math-related datasets, namely GSM8K (Cobbe et al., 2021) and MATH (Hendrycks et al., 2021), because the broad community interest in mathematical reasoning has led to the development of a large, diverse set of teachers that are readily available and suitable for distillation. We train LLaMA-1B-Base, OLMo-1B-Base and Gemma-2B-Base (for GSM8K) as well as LLaMA-3B-Base (for MATH) using generations sampled from 15 candidate teachers drawn from the LLaMA (Team, 2024c), OLMo (OLMo, 2024), Qwen (Qwen et al., 2025), Gemma (Team, 2024b), and Phi (Abdin et al., 2024) families. Our results show that:

- GRACE correlates strongly with the student's distillation performance (Figure 1, left), outperforming strong baselines such as G-Vendi (Jung et al., 2025).
- GRACE identifies teachers that are strongly compatible to a given student. Selecting teachers using GRACE yields 7% and 2% improvement in student performance compared to using the best-performing teacher, on GSM8K and MATH respectively.
- GRACE offers actionable insights to practitioners. It helps identify close to 1) optimal generation temperature for a given teacher model, 2) best teacher within a size constraint, and 3) best teacher within a model family.

Given the limited research on teacher selection in LLM distillation, our findings indicate that GRACE improves performance and offers reliable guidance for effective distillation.

## 2 GRACE: GRADIENT CROSS-VALIDATION EVALUATION

We consider the case of using distillation to fine-tune a pre-trained student model to solve specific downstream tasks. For each of the $N$ prompts $\mathbf{x} \in \mathcal{X}$, we autoregressively generate $M$ responses $y_1, ..., y_M$ from a teacher distribution $\pi_{\mathcal{T}}$. This distribution encodes the temperature it may be

sampled at from the teacher as well. We then fine-tune the pre-trained student with the standard autoregressive cross-entropy objective $\mathcal{L}$ on a dataset $\mathcal{D}_{\mathcal{T}}^{\text{distill}}$ containing $N \times M$ teacher generations. In contrast to logit-based distillation, learning from generations permits distillation across architectures and in cases where the teacher's logits are not available. We measure the performance of students and teachers as the average accuracy of $k$ sampled responses for a given prompt (i.e., average-at-k). We will use $\pi_{\mathcal{S}}$ to denote the pre-trained student, and refer to its parameters as $\Theta_{\mathcal{S}} \in \mathbb{R}^D$ when necessary.

## 2.1 GRADIENT-BASED SCORES

The problem of selecting a teacher for distillation is closely connected to the well-studied field of data selection: choosing the best teacher based on its generations can be viewed as selecting the best subset from the union of all teachers' generations, with the constraint that each subset must come from a single teacher. For language models, many successful data selection methods (Pruthi et al., 2020; Xia et al., 2024; Engstrom et al., 2024) rely on first- or second-order gradient information to identify useful data for a given task.

In contrast to selecting individual datapoints from a dataset, we would like to select a data distribution (i.e., corresponding to a teacher). As such, instead of quantifying the value of individual datapoints, we turn our attention to gradient-based approaches to measure data quality in terms of its *distributional features*. For a teacher $\pi_{\mathcal{T}}$, we assume access to only a subsampled dataset $\mathcal{D}_{\mathcal{T}}^{\text{eval}} \subset \mathcal{D}_{\mathcal{T}}^{\text{distill}}$ containing $n \times m$ prompt-generation pairs, where $n, m$ may be much smaller than $N, M$. In our experiments (Section 3), $n \times m$ is $60\times$ smaller compared to the $N \times M$.

In the following, we will first describe two baselines (i.e., G-Vendi (Jung et al., 2025) and G-Norm), then introduce the proposed GRACE score combining the advantages of both. The two baselines measure data diversity (via directional entropy) and teacher-student alignment (via gradient norms). GRACE constructs a single score that balances these two desiderata by penalizing large gradients along low-eigenvalue directions.

**Gradients.** We establish some useful notation to work with gradients. Let $\boldsymbol{g}(\mathbf{x}, \mathbf{y}) := \frac{1}{|\mathbf{y}|}\nabla\mathcal{L}(\mathbf{y}|\mathbf{x}; \Theta_{\mathcal{S}})$ be the student's gradient on the response $\mathbf{y}$ conditioned on prompt $\mathbf{x}$, which is averaged over response tokens, following prior works on data selection. Since all gradients are computed with respect to the student model's parameters, we omit the explicit dependency on $\Theta_{\mathcal{S}}$ for notational clarity.

We process the gradient with two steps. First, for computational reasons, we work with a random low-dimensional projection of the gradient, denoted $\Pi\boldsymbol{g} \in \mathbb{R}^d$ with $\Pi \in \{\pm 1/\sqrt{D}\}^{d \times D}$ (Park et al., 2023). We also rescale the gradient by $\log|\mathbf{y}|$ to account for the response length $|\mathbf{y}|$. This is motivated by the empirical observation that the gradient norm averaged over a length-$T$ sequence roughly decreases as $1/\log T$ (Figure 28), which can cause gradient-based computations to unduly favor short sequences (Xia et al., 2024).

The processed gradient is denoted $\boldsymbol{h}(\mathbf{x}, \mathbf{y}) := \log(|\mathbf{y}|) \cdot \Pi\boldsymbol{g}(\mathbf{x}, \mathbf{y})$. For a dataset $\mathcal{D}$ of generations, we define the matrix consisting of processed gradients (i.e., $\boldsymbol{h}$) as $\boldsymbol{G}(\mathcal{D}) \in \mathbb{R}^{nm \times d}$, and define the matrix consisting of processed and *normalized* gradients (i.e., $\tilde{\boldsymbol{h}} = \boldsymbol{h}/\|\boldsymbol{h}\|$) as $\tilde{\boldsymbol{G}}(\mathcal{D}) \in \mathbb{R}^{nm \times d}$. Then, we define the mean and second moment of the unnormalized gradients as:

$$\mu(\mathcal{D}) := \frac{1}{nm}\boldsymbol{G}(\mathcal{D})^{\top}\mathbf{1}, \quad \boldsymbol{\Sigma}(\mathcal{D}) := \frac{1}{nm}\boldsymbol{G}(\mathcal{D})^{\top}\boldsymbol{G}(\mathcal{D}). \tag{1}$$

$\tilde{\mu}(\mathcal{D}), \tilde{\boldsymbol{\Sigma}}(\mathcal{D})$ are defined similarly. We additionally use $\hat{\boldsymbol{\Sigma}}(\mathcal{D}) = \tilde{\boldsymbol{\Sigma}}(\mathcal{D}) + \frac{\nu}{d}\boldsymbol{I}$ with smoothing parameter $\nu > 0$ for numerical stability.

Gradient information has been used for data selection, where samples with higher influence to a validation set is preferred (Pruthi et al., 2020; Xia et al., 2024). One can adapt these methods for teacher selection. However, we find them to be not strongly predictive empirically; we provide details in Section C.1. Consequently, we evaluate training data quality by analyzing its distributional properties, without relying on a validation set.

**G-Vendi (Jung et al., 2025).** One natural distributional measure of data quality is diversity. Along these lines, Jung et al. (2025) propose the G-Vendi score, which measures the directional coverage of

$\mathcal{D}$ as the entropy of the eigenvalues of $\tilde{\boldsymbol{\Sigma}}(\mathcal{D})$ (Equation (1)):

$$\text{G-Vendi}(\mathcal{D}) := \text{Entropy}(\lambda(\tilde{\boldsymbol{\Sigma}}(\mathcal{D}))) = - \sum_{\lambda \in \lambda(\tilde{\boldsymbol{\Sigma}}(\mathcal{D}))} \lambda \log \lambda, \tag{2}$$

where $\lambda(\tilde{\boldsymbol{\Sigma}}(\mathcal{D}))$ denotes the set of eigenvalues of $\tilde{\boldsymbol{\Sigma}}(\mathcal{D})$, with $|\lambda(\tilde{\boldsymbol{\Sigma}}(\mathcal{D}))| = \min\{nm, d\}$. A larger G-Vendi score is preferred as it corresponds to better diversity. Jung et al. (2025) use G-Vendi to select an optimal subset of training data $\mathcal{D}$ from a full dataset generated by a single teacher. However, using G-Vendi to select a teacher out of many candidates may yield suboptimal choices. For example, when the student serves as its own teacher, we find that its G-Vendi score (5.93) is higher than that of all other teachers (Figure 2), even though the self-distilled student has a low accuracy of 4%. This occurs because an untrained model can produce random responses, which exhibit high gradient entropy and therefore a high G-Vendi score, yet offer no meaningful learning signal for distillation.

**G-Norm.** The limitation of G-Vendi leads us to investigate another gradient-based distributional score, namely the gradient-norm score (G-Norm):

$$\text{G-Norm}(\mathcal{D}) := \text{Tr}\left(\boldsymbol{\Sigma}(\mathcal{D})\right) = \frac{1}{nm} \sum_{(\mathbf{x},\mathbf{y}) \in \mathcal{D}} \|\boldsymbol{h}(\mathbf{x},\mathbf{y})\|^2. \tag{3}$$

Because gradients become smaller near an optimum, a lower G-Norm indicates that the student requires less parameter updates to reduce loss on the given task. G-Norm therefore serves as a proxy for teacher-student *alignment*. This explains why high-performing models may be poor teachers: although Gemma-2 Instruct models achieve strong accuracy on GSM8K and MATH, the student model shows high G-Norm on their generated responses suggesting weak teacher-student alignment and shows weaker post-distillation performance (Figure 2). For students which have been pretrained, G-Norm can additionally reflect the teacher's generation *quality*, as G-Norm is high when a teacher provides uninformative or random responses. In particular, using the student itself as the teacher results in the highest G-Norm, showing that its outputs do not provide a meaningful distillation signal.

Nevertheless, G-Norm alone remains insufficient. For instance, on GSM8K, we find that G-Norm's value largely differentiates different teacher families and does not correlate with the student's performance when generation temperature from the teacher is varied (Figures 2 and 5). This is because G-Norm captures only the magnitude of gradients, without considering the directional spread of the gradients across dimensions.

G-Norm and G-Vendi capture complementary distributional properties and can sometimes trend in different directions. For instance, we find that increasing the teacher's generation temperature increases G-Norm, consistent with the observation that higher temperatures induce worse data, but at the same time increases G-Vendi, indicating higher data diversity (Figure 5). As such, we treat G-Norm and G-Vendi as baselines and propose GRACE as a single score that unifies both.

## 2.2 THE GRACE SCORE

GRAdient Cross-validation Evaluation (GRACE) computes the norm of the gradients weighted under the spectrum of the normalized gradient second-moment matrix, thereby combining the advantages of G-Vendi and G-Norm. GRACE is computed solely using the student's gradients on the teacher's generations, and does not rely on a verifier or access to test samples. In the following, we will first define the GRACE score, and then describe its connection to leave-one-out conditional mutual information.

**GRACE.** For a dataset $\mathcal{D}$ of teacher generations containing $n \times m$ prompt-generation pairs and a choice of hyperparameter $C$, construct $C$ partitions of the prompts in the dataset $\mathcal{D}$, denoted $\{\mathcal{D}_i\}_{i=1}^C$, each containing $n/C$ prompts and their generations. Let $\mathcal{D}_{-i}$ denote the concatenation of all partitions except the partition $\mathcal{D}_i$. Then, GRACE is defined as

$$\text{GRACE}(\mathcal{D}) = \frac{1}{C} \sum_{i=1}^C \text{Tr}\left(\hat{\boldsymbol{\Sigma}}(\mathcal{D}_{-i})^{-1}\boldsymbol{\Sigma}(\mathcal{D}_i)\right) = \frac{1}{nm} \sum_{i=1}^C \sum_{(\mathbf{x},\mathbf{y}) \in \mathcal{D}_i} \|\hat{\boldsymbol{\Sigma}}(\mathcal{D}_{-i})^{-1/2}\boldsymbol{h}(\mathbf{x},\mathbf{y})\|^2, \tag{4}$$

where $\hat{\boldsymbol{\Sigma}}(\mathcal{D}_{-i}) = \tilde{\boldsymbol{\Sigma}}(\mathcal{D}_{-i}) + \frac{\nu}{d}\boldsymbol{I}$ with smoothing parameter $\nu > 0$ for numerical stability.

GRACE computes the expected squared norm of spectral-weighted gradients. Given a random partition $(\mathcal{D}_i, \mathcal{D}_{-i})$, let $\{\lambda_j, \boldsymbol{u}_j\}_{j \in [d]}$ denote the set of eigenvalues and eigenvectors for $\tilde{\boldsymbol{\Sigma}}(\mathcal{D}_{-i})$. For this partition, GRACE computes

$$\sum_{j \in [d]} \frac{1}{\lambda_j + \frac{\nu}{d}} \left( \frac{1}{|\mathcal{D}_i|} \sum_{(\mathbf{x}, \mathbf{y}) \in \mathcal{D}_i} (\boldsymbol{h}(\mathbf{x}, \mathbf{y})^\top \boldsymbol{u}_j)^2 \right). \tag{5}$$

**A bias-variance tradeoff.** GRACE can be split into two terms:

$$\text{GRACE}(\mathcal{D}) \approx \underbrace{\frac{1}{nm} \sum_{i=1}^{C} \text{Tr}\left( \hat{\boldsymbol{\Sigma}}(\mathcal{D}_{-i})^{-1} \boldsymbol{G}_\mu(\mathcal{D}_i)^\top \boldsymbol{G}_\mu(\mathcal{D}_i) \right)}_{\text{GRACE-Variance}(\mathcal{D})} + \underbrace{\frac{1}{nm} \sum_{i=1}^{C} \mu(\mathcal{D})^\top \hat{\boldsymbol{\Sigma}}(\mathcal{D}_{-i})^{-1} \mu(\mathcal{D})}_{\text{GRACE-Bias}(\mathcal{D})},$$

where $\boldsymbol{G}_\mu(\mathcal{D}) := \boldsymbol{G}(\mathcal{D}) - \mathbf{1}\mu(\mathcal{D})^\top$ denotes the centered gradient matrix. The term GRACE-Variance$(\mathcal{D})$ captures the variance in gradients under the spectrum of $\hat{\boldsymbol{\Sigma}}$, while GRACE-Bias$(\mathcal{D})$ measures the norm of the mean gradient under the spectrum of $\hat{\boldsymbol{\Sigma}}$.

Repeating our intuitions from G-Norm, GRACE-Bias is highly useful for identifying pathological teachers: for example, if a teacher provides random responses, GRACE-Bias becomes large, indicating that such data is not suitable for distillation.

When we do not encounter such teachers, most of the predictive power comes from GRACE-Variance. In our experiments, GRACE-Variance dominates GRACE-Bias, and our conclusions remain the same whether we consider GRACE or GRACE-Variance. A smaller GRACE-Variance indicates a better distillation teacher, as it corresponds to a smaller variance of gradients along all eigenvectors of $\hat{\boldsymbol{\Sigma}}$. Gradient variance in directions with smaller eigenvalues are penalized more heavily, which is desirable as high gradient variance along such directions can more easily induce instability or poor generalization. We formalize this view in Section 2.2.1. The directional spectrum is taken from the *normalized* gradients, since the gradient norm is less relevant than the gradient direction with the use of adaptive optimizers and normalization layers (Loshchilov & Hutter, 2019; Ba et al., 2016; Li et al., 2022).

### 2.2.1 CONNECTING GRACE TO LEAVE-ONE-OUT CMI

GRACE connects naturally to leave-one-out conditional mutual information (CMI), a frequently used concept in studying generalization (Xu & Raginsky, 2017; Steinke & Zakynthinou, 2020; Haghifam et al., 2022; Rammal et al., 2022). CMI captures how sensitive the learning outcome is to the removal of a single sample. A higher sensitivity suggests heavier memorization, which can lead to low generalization to unseen test examples. Under this framework, we show that GRACE successfully unifies G-Norm and G-Vendi.

Formally, we overload $\boldsymbol{g}(\mathcal{D}; \Theta) = \frac{1}{|\mathcal{D}|} \sum_{(\mathbf{x}, \mathbf{y}) \in \mathcal{D}} \boldsymbol{g}(\mathbf{x}, \mathbf{y}; \Theta)$ to denote the average gradient on a dataset $\mathcal{D}$. To keep our discussion general, we consider the following gradient update that uses gradients and a preconditioner matrix $\mathbf{M}$:

$$\Theta \leftarrow \Theta - \eta(\mathbf{M}(\mathcal{D}; \Theta)\boldsymbol{g}(\mathcal{D}; \Theta) + \epsilon),$$

where $\eta$ denotes the learning rate and $\epsilon \sim \mathcal{N}(0, \sigma^2 \mathbf{I})$ denotes the gradient noise. Setting $\mathbf{M}$ as identity recovers gradient descent, and setting $\mathbf{M}$ as a function of gradient second moments recovers various adaptive algorithms in practice.

Let $\Theta'_{\mathcal{D}}$ denote the resulting parameters after a gradient update with $\mathcal{D}$, and $\Theta'_{\mathcal{D} \setminus \{(\mathbf{x}, \cdot)\}}$ denote the parameters from a set where all training data connected to a uniformly sampled prompt $\mathbf{x}$ are dropped from the training set $\mathcal{D}$. CMI measures the mutual information between the parameters $\Theta'_{\mathcal{D} \setminus \{(\mathbf{x}, \cdot)\}}$ and the dropped prompt $\mathbf{x}$. We show that CMI can be bounded as follows:

**Lemma 1** (Informal; cf Theorem D.2). *Define GRACE with $C = n$. For any $\mathcal{D}'$, take $\mathbf{M}(\mathcal{D}', \Theta) := \hat{\boldsymbol{\Sigma}}(\mathcal{D}')^{-1/2}$, where then CMI $\lesssim \frac{1}{\sigma^2 n^2}$ GRACE-Variance $(\mathcal{D}) \lesssim \frac{1}{\sigma^2 n^2}$ GRACE $(\mathcal{D})$.*

Thus, GRACE offers an upper bound on the student's generalization performance after training with a single gradient update under the preconditioner matrix $\hat{\boldsymbol{\Sigma}}^{-1/2}$. Intuitively, GRACE captures how

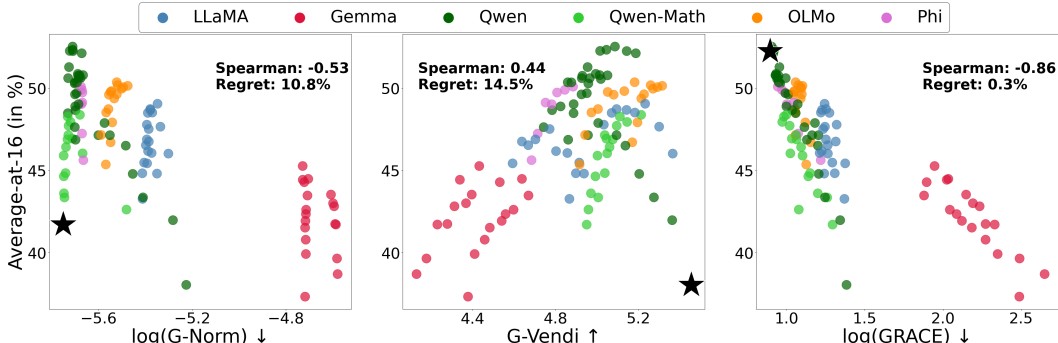

Figure 2: **GRACE achieves** $86\%$ **Spearman correlation to LLaMA-1B's post-distillation performance on GSM8K,** much higher than G-Norm (53%) and G-Vendi (44%). When evaluated by teacher-student regret, GRACE selects a teacher with regret of $0.3\%$, outperforming G-Norm and G-Vendi, which incur regrets of $10.8\%$ and $14.5\%$, respectively. Stars denote students trained from the teacher chosen by each score. Gemma teachers are outliers, because they give extremely concise responses to each prompt. More discussion is in Section G.1

uniformly spread the gradients are across examples, which mirrors the core idea behind CMI. Higher gradient uniformity leads to more stable optimization, which in turn can help stronger performance on held-out evaluation.

There are several ways to extend Lemma 1. First, Lemma 1 is based on a particular choice of the preconditioner matrix (i.e. $\mathbf{M}(\mathcal{D}', \Theta) = \hat{\mathbf{\Sigma}}(\mathcal{D}')^{-1/2}$), which is motivated by adaptive optimizers used in practice (Kingma, 2014; Loshchilov & Hutter, 2019; Duchi et al., 2011). In principle, one could obtain even tighter upper bounds on CMI by choosing $\mathbf{M}$ optimally. Further, one could improve Lemma 1 by connecting directly to performance metrics other than the loss, and by considering multiple gradient steps. We provide more discussions in Section D.

## 3 EXPERIMENTS

We compare the three scores mentioned in the previous section, G-Norm, G-Vendi, and GRACE, on two common math reasoning datasets, GSM8K (Cobbe et al., 2021) and MATH (Hendrycks et al., 2021). These datasets have a diverse set of strong teacher models readily available, due to the broad community interest in mathematical reasoning. For each prompt-response pair, the model receives a binary correctness score, and we quantify its performance by the average accuracy (in percentage) achieved when sampling $k$ responses for each prompt, referred to as average-at-$k$.

**Settings.** The student model is taken to be LLaMA-1B-base, OLMo-1B-base or Gemma-2B-base on GSM8K (Cobbe et al., 2021), and LLaMA-3B-base on MATH (Hendrycks et al., 2021). We compare 15 text-only trained teachers: LLaMA-(3.2/3.3) 3/8/70B Instruct models, Qwen-2.5 1.5/3/7/14B Instruct models, Qwen-2.5 Math 1.5/7B Instruct models, Gemma-2 2/9/27B Instruct models, OLMo 7/13B Instruct models, and Phi-4 on both MATH and GSM8K (Dubey et al., 2024; Abdin et al., 2024; Yang et al., 2024; Qwen et al., 2025; Team, 2024a).[1] The teacher's generation temperature is varied from 0.3 to 1.0 in increments of 0.1. The computational costs of GRACE are detailed in Section E.7. We also compare against several common baselines, described in detail in Section C.3. All baselines considered do not require access to a verifier except for teacher performance.

To compute scores, we use a subset of $n = 512$ prompts randomly selected from the training set, with $m = 4$ generations per prompt. For GRACE, we use 10-way cross validation (i.e. $C = 10$). The student gradients are randomly projected to dimension $d = n = 512$; we provide ablation results on these hyperparameter choices in Section 3.3.

---

[1]We use short chain-of-thought reasoning models as teachers, given the extensive number of such available models, which also enables extensive ablations. Our experiments focus on small-scale students, for which prior work has shown that long chain-of-thought teachers do not necessarily benefit such models (Li et al., 2025).

Each distillation training run uses learning rate[2] $10^{-5}$ and 4 epochs over the training set. We use the cosine learning rate schedule with $5\%$ warmup, 0 weight decay, and batch size 64. We generate $M = 16$ responses per prompt from each teacher and fine-tune the student on all generations without filtering for correctness of the final answer.[3]

**Evaluation.** We evaluate the utility of each score with two tests. First, we compute the Spearman correlation of each score with the performance of the trained student models. Second, we measure the *teacher-student regret*, defined as the difference between the final student performance when trained by the actual best teacher, and the final student performance when trained under the best teacher selected by each score. Performances of the best students are reported in Table 3.

We measure performance of the student by average-at-16 performance based on 16 generations at temperature 1.0. Average-at-16 is a more stringent evaluation metric that tests consistency of generation from the model by requiring to solve a problem correctly across multiple attempts. It has recently been used for assessing robust performance on olympiad-level benchmarks and also serves as a key loss component when training models via reinforcement learning (Guo et al., 2025). We discuss later in Section 3.3 how the results change when we look at other performance metrics.

**Scatter plots.** We present our findings in form of scatter plots (e.g. Figures 2 and 10). In all scatter plots, each scatter point represents the student's distillation performance from a specific teacher–temperature pair. Different teacher families are shown in distinct colors for clarity. We mark best teacher selected by each score by "star" in its corresponding subplot.

## 3.1 GRACE CORRELATES WELL WITH STUDENT'S PERFORMANCE AND RETURNS LOW REGRET

Figure 2 shows that for a LLaMA-1B model trained on GSM8K, GRACE achieves the best Spearman correlation with the student performance (0.86) when compared against G-Norm (0.55) and G-Vendi (0.44). Furthermore, GRACE returns a lower regret (0.3 %) when selecting the best teacher, compared to G-Norm (4.9 %) and G-Vendi (14.5%) respectively.

Additional experiments with OLMo-1B and Gemma-2B models trained on GSM8K (Appendix Figures 10 and 11) as well as with a LLaMA-3B model trained on MATH (appendix Figure 14) verify the utility of GRACE. Summary of all our findings is in Table 3 in the appendix. We also compare against other data selection baselines in Figure 3. Across both GSM8K and MATH, GRACE stands out as the only score maintaining a strong correlation ($> 85\%$) with student performance, and simultaneously yields the lowest regret, $0.3\%$ on GSM8K and $3.9\%$ on MATH.

**Comparisons with teacher performance and student loss.** Two intuitive baselines fail to reflect the student's performance. The first is the teacher's own performance, which only shows a weak correlation of $11\%$ for LLaMA-1B on GSM8K. This agrees with findings in prior works (Mirzadeh et al., 2019; Harutyunyan et al., 2023; Panigrahi et al., 2025; Zhang et al., 2023; 2025a; Peng et al., 2025; Razin et al., 2025). As an example, LLaMA-70B Instruct has the best performance among all teachers, but a student trained with LLaMA-70B Instruct reaches only $44.5\%$ average-at-16 performance, which returns a regret of $7.7\%$ when compared against the best performing student. Similarly, the student's loss on teacher's generations, measured before training, is also poorly correlated with the student's performance ($44\%$ with LLaMA-1B training on GSM8K) and returns a regret of $5.4\%$.

## 3.2 GUIDING DISTILLATION PRACTICE WITH GRACE

GRACE can go beyond identifying the best teacher and inform distillation practices. Below we discuss how GRACE provides guidance under common scenarios.

**Selecting generation temperature.** The temperature $\tau$ used to rescale the teacher's logits when generating responses is known to have a strong influence on student performance after distillation (Zheng & YANG, 2024; Peng et al., 2025). However, there hasn't been a principled approach to choose the temperature. We show in Figure 5 that GRACE can identify such a good generation temperature for two Qwen teachers: it closely predicts the optimal generation temperature for LLaMA-1B training, which are 0.8 (vs. predicted 0.9) with the 3B teacher and 0.4 (vs. predicted 0.5) with the 1.5B

---

[2]We search over learning rates $\{5 \times 10^{-5}, 10^{-5}, 5 \times 10^{-6}\}$ and find $10^{-5}$ to be consistently the best.

[3]Surprisingly, ablations in Section F.1 show that results are not significantly affected if we filter by correctness.

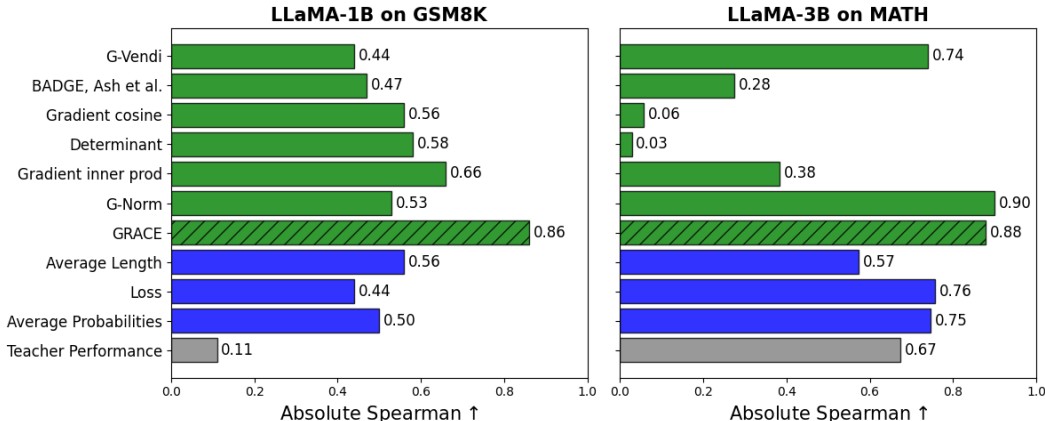

Figure 3: **GRACE achieves the strongest correlation to student performance**, among all scores for LLaMA-1B on GSM8K and LLaMA-3B on MATH. Blue bars represent gradient-based scores, green bars denote student logit-based scores on the training data, and the gray bar corresponds to teacher performance. Teacher performance and the student's loss on teacher generations (Loss) before training show only weak correlations. While G-Norm correlates well with student performance on MATH, it is significantly worse on GSM8K.

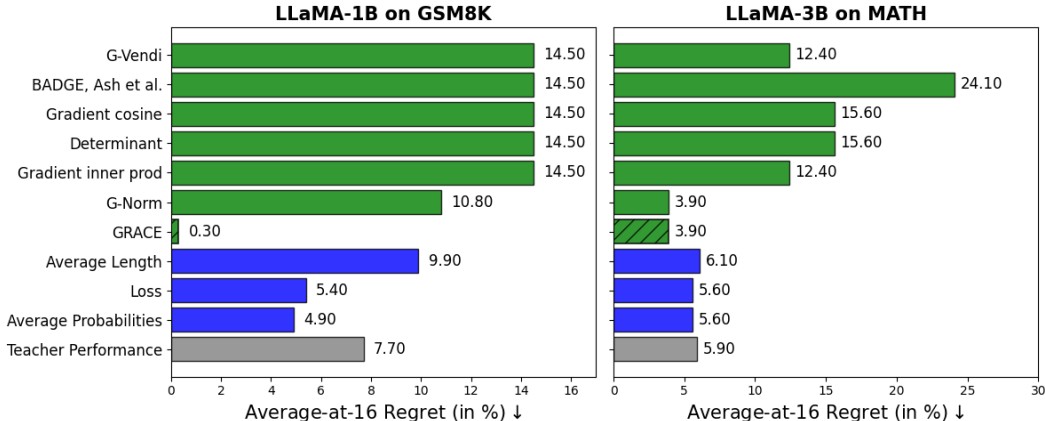

Figure 4: **GRACE achieves the minimum regret**, among all scores for LLaMA-1B on GSM8K and LLaMA-3B on MATH. Naively selecting the teacher with the best performance shows a regret of at least $7.7\%$ and $5.9\%$ in performance of student training on GSM8K and MATH respectively. On the other hand, GRACE achieves a regret of $0.3\%$ and $3.9\%$ on GSM8K and MATH respectively. In the left plot, scores that show $14.5$ regret select the same teacher, resulting in identical regret values.

teacher. In comparison, G-Norm and G-Vendi tend to increase monotonically with the temperature, even though the student's performance shows an inverse U-shape in temperature. Aggregating across all teachers in Figure 5 (right), GRACE achieves a 75% correlation with student performance as temperature varies, substantially outperforming G-Vendi (59%). Interestingly, G-Norm is negatively correlated ($-53\%$) with performance in this setting.

**Selecting a teacher within a size budget**. In practice, one common resource constraint for distillation is the compute required to locally host open-source teachers. Motivated by this, we test whether GRACE can be used to select a teacher under a given size. Specifically, we evaluate three scale constraints: (1) 3B and below, (2) 10B and below, and (3) 30B and below. Figure 6 shows that GRACE achieves over 79% correlation with student performance across all scale constraints. Moreover, within each scale constraint, using GRACE to select a teacher yields a regret below $0.3\%$, compared to at least 9% regret with G-Norm and G-Vendi.

**Selecting teachers within a model family.** Another practical limitation is the family of models that one can access, motivating us to test GRACE against models within each model family. We split

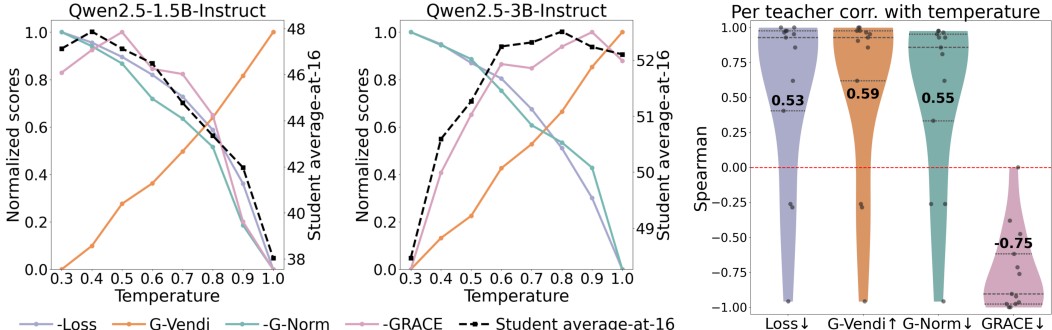

Figure 5: **GRACE identifies a good teacher generation temperature.** (Left) Results are shown for LLaMA-1B trained with Qwen-2.5-1.5B-Instruct and Qwen-2.5-3B-Instruct teachers on GSM8K. GRACE identifies (1) a lower temperature is optimal for Qwen-2.5-1.5B-Instruct, and (2) a higher temperature is effective for Qwen-2.5-3B-Instruct. In contrast, G-Norm can only identify (1) and G-Vendi can only identify (2). For clarity, all scores are normalized to [0, 1]. The signs of Loss, G-Norm, and GRACE are inverted so that all scores become higher-is-better for better visualization. (Right) Average correlations of student performance with each score when the generation temperature is varied from a teacher. GRACE achieves 75% correlation, higher than G-Vendi (59%) and other scores.

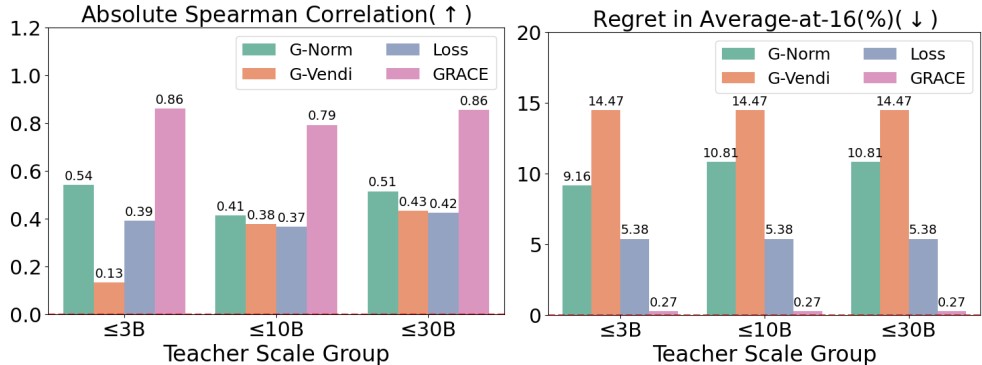

Figure 6: **GRACE effectively predicts student performance under scale-constrained teachers.** Results are for LLaMA-1B on GSM8K. (Left) Across all teacher scale constraints, GRACE has high correlation (at least 79%) with the student performance after training with the teacher. (Right) Across all teacher scale constraints, GRACE-selected teacher has minimum regret ($< 1\%$) when compared against other scores. In the right plot, G-Vendi and Loss consistently select the same teacher across all scale constraints, resulting in identical regret values in each case.

the teacher models by model family and consider all generation temperatures. Since some families include only a small number of teachers, the Spearman correlations can be unreliable. We hence report the regret of GRACE for each teacher family. As shown in Figure 15 in appendix, GRACE achieves an average regret of just 1% when selecting a teacher within each family, while other metrics yield at least 3% regret on average. Interestingly, the best teacher is not always from the same family as the student. For instance, a LLaMA-1B student learns better from a Qwen-Instruct teacher than from any LLaMA-Instruct teacher.

## 3.3 ABLATIONS AND ADDITIONAL EXPERIMENTS

**Effect of hyperparameters.** We test the effect of various hyperparameters used in the GRACE computation. We vary the number of prompts ($n$), the number of generations per prompt ($m$), and the dimension of the gradient random projection ($d$). For the LLaMA-1B student on GSM8K, we find that GRACE is generally robust to these hyperparameter choices, and the default values ($m = d = 512$, $m = 4$) work well (see details in Section F.3). We also vary the number of cross-validation splits used in GRACE. For both GSM8K and MATH, the correlation with student performance remains fairly stable once $C >= 6$ (Figure 27), so we set $C = 10$ for our experiments.

**Robustness of the scores.** To test the robustness with respect to teacher selection, we evaluate correlations on random subsets of teachers. In addition to the case studies in Section 3.2, we repeatedly compute scores over random subsets of teachers. As shown in Figure 29, GRACE consistently maintains high correlations across these subsets (see details in Section F.5).

**Changing Average-at-k to other performance measures.** We further examine how correlations change when replacing Average-at-k with other evaluation metrics. For GSM8K, we find that Spearman correlation drops when switching from Average-at-k to either greedy or best-of-k accuracy, but GRACE still returns the lowest regret in the corresponding performance measure ($< 2.5\%$) when selecting the best teacher (Tables 5 and 6 in appendix). Greedy reflects performance from a single generation at temperature $0.0$, and best-of-k measures whether the student answers correctly at least once over $k$ responses at generation temperature $1.0$. A deeper investigation into the discrepancy between Average-at-k and these discrete performance metrics is left to future work.

**Out-of-distribution evaluation:** We evaluate LLaMA-1B trained on GSM8K on GSM-Symbolic (Mirzadeh et al., 2025) (Figure 16), and LLaMA-3B trained on MATH on MATH$^2$ (Shah et al., 2024) and MATH Perturb (Huang et al., 2025b) (Table 4). On GSM8K, student performance is strongly correlated between GSM8K and GSM-Symbolic, and GRACE is highly predictive under this OOD shift.

For MATH, we observe a nuanced pattern. On the simpler OOD task MATH Perturb (Simple), G-Norm is a stronger predictor of student performance across teachers than GRACE. However, on the more challenging OOD settings, MATH$^2$ and MATH Perturb (Hard), GRACE achieves similar or higher spearman correlation and lower regret than G-Norm, indicating that GRACE is a reliable score for identifying teachers for which the trained student generalizes well on difficult OOD tasks.

## 4 DISCUSSION

This work proposes the GRACE score for identifying the most suitable distillation teacher. Motivated from optimization and connected to CMI, GRACE leverages two distributional properties of the student's gradients: the directional coverage of the (normalized) gradients, and the gradient variance. Experiments on GSM8K and MATH establish that GRACE strongly correlates with the student's performance after distillation, and enables principled comparison across teachers and offers actionable insights into practical scenarios. Our results highlight GRACE's potential as a practical and general-purpose tool for guiding distillation practices.

Below we discuss promising avenues for future work.

First, GRACE is not guaranteed to be robust in adversarial setups. One example is when a teacher repeatedly emits partial reasoning steps or redundant text before eventually arriving at the correct answer. Due to the in-context ability of language models, growing the number of repetitions will make the prediction closer to deterministic, and hence having an increasingly small gradient. In this case, the GRACE score may be deceptively low even though the student can perform arbitrarily poorly. Fortunately, such behaviors are rare among well-performing teachers and can be mitigated via post-processing. Fully characterizing the failure cases of GRACE remains an open question.

Moreover, while GRACE captures two important distributional properties of the student's gradients, incorporating properties of the teacher and characteristics of the data can be fruitful. For example, although GRACE's design intentionally avoids requiring teacher logits, incorporating logit-level information where available may lead to further performance gains. It is also interesting to investigate GRACE's utility in adaptive distillation strategies, where teacher choice may vary dynamically across training stages or subsets of data, possibly in an online fashion akin to reinforcement learning.

Finally, while this work focuses on mathematical reasoning due to the abundance of teacher choices and the ease of verification, GRACE is not defined specifically for math. Extending GRACE to more general domains would be an interesting direction for future work. As a preliminary study, we extend our analysis to the ARC dataset (Clark et al., 2018) in Table 7. GRACE achieves the highest Spearman correlation with student performance among all scores that do not rely on a verifier, and achieves the lowest regret among all scores. Interestingly on this task, we observe cases where teachers with extremely high GRACE scores can still yield strong students. Understanding the broader conditions under which GRACE reliably predicts student performance is a promising future direction.

ACKNOWLEDGEMENTS

We thank Jordan Ash, Alex Damian, Tianyu Gao, Jonathan Geuter, Sara Kangaslahti, Nihal Nayak, Xingyu Zhu, Simon Park, and Noam Razin for helpful discussions and suggestions. Abhishek Panigrahi acknowledges support from Apple AIML and Siebel Scholarships. This work was enabled in part by a gift from the Chan Zuckerberg Initiative Foundation to establish the Kempner Institute for the Study of Natural and Artificial Intelligence.

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

CONTENTS

## A  OUTLINE OF THE APPENDIX

Below, we outline the structure of the appendix.

Section B provides situates our work among existing literature. Section C details the baseline methods against which we compare GRACE. Section C.1 introduces gradient influence–based scores and explains why they are ineffective for identifying the best teacher. Additional scoring methods are discussed in Section C.3.

Section D formally defines conditional mutual information (CMI), which provides the theoretical basis for GRACE. Section D.3 highlights key limitations of using CMI to explain the success of GRACE and outlines open questions to bridge this gap.

Section E shows additional experimental results: additional models on GSM8K and MATH (Section E.1); results within individual teacher families (Section E.2); the effect of teacher performance on GRACE (Section E.4); correlations for metrics beyond Average-at-16 (Section E.5); evaluation on non-math datasets (Section E.6); and computational complexity analysis (Section E.7). Finally, Section F presents ablations studying key design choices in GRACE.

## B  RELATED WORK

**Knowledge distillation.**  Knowledge distillation is a classic method used to improve the optimization and generalization of a small model (Hinton et al., 2015). A counterintuitive finding is that a better-performing model is not necessarily a better teacher, which has been observed in both classic classification or regression settings (Mirzadeh et al., 2019; Jafari et al., 2021; Harutyunyan et al., 2023) and more recently in language models (Zhang et al., 2025a; 2023; Xu et al., 2025; Panigrahi et al., 2025). For language models, one can distill from either the logits of the teacher or the generated texts. [4] While the former can lead to better student performance, it is more computationally costly, requires higher access, and is less flexible due to tokenizer choices. We hence focus on distilling from generated texts (Eldan & Li, 2023; Li et al., 2023; Busbridge et al., 2025). Recent work by Guha et al. (2026) supports our findings: they demonstrate that a weaker teacher can yield a stronger distilled model, that distillation benefits from increased sample size, and that filtering has little impact on the resulting student's performance.

**Data selection.**  For text-based distillation, selecting the best teacher can be considered as the problem of choosing the most useful subset of samples from the generations of all teachers. This aligns with the broad task of *data selection*, which aims to identify subsets of data that maximize certain utility (Sorscher et al., 2022; Albalak et al., 2024). Many approaches leverage gradient

---

[4]We consider generations following standard next-token distributions, as opposed to antidistillation sampling (Savani et al., 2025).

information (Mirzasoleiman et al., 2020; Killamsetty et al., 2021; Pruthi et al., 2020; Xia et al., 2024), including some that directly rely on notions of coverage (Ash et al., 2019; Jung et al., 2025). Directional coverage also ties to the notion of coverage in reinforcement learning. Specifically, autoregressive training on teacher generations can be viewed as a form of behavior cloning, for which increasing the coverage is provably beneficial (Song et al., 2024; Huang et al., 2025a; Rohatgi et al., 2025). Despite these similarities, distillation differs from standard data selection in that it allows generating new data and offers a richer design space (Peng et al., 2025). An effective teacher-selection score should therefore be versatile and broadly applicable across scenarios, a property that GRACE demonstrates as shown in Section 3.2.

Zhang et al. (2025b) study teacher selection at the granularity of individual prompts, proposing to score a pool of responses from different teachers by the student's loss and train on the best-scoring completion for each prompt. However, as also observed by the contemporary work (Just et al., 2025) and confirmed in our experiments, such loss-based scoring is only a coarse proxy for response quality. Just et al. (2025) propose using loss on short reasoning segments to mitigate this. They show that such scoring can be used to select the best global teacher, however their evaluation is limited to only three teachers. By contrast, we evaluate GRACE across fifteen teachers and eight temperature settings each. Extending GRACE to per-prompt teacher selection is a promising direction for future work.

## C  DISCUSSION ON THE BASELINES

### C.1  USING GRADIENT INFLUENCE FOR TEACHER SELECTION

Existing data selection methods estimate the importance of each data point based on its influence on a validation set. Among these, gradient-based approaches define this influence through the relationship between the gradient of an individual training example and the gradients computed on the validation set.

A straightforward extension of these data selection strategies is to measure the quality of the gradient distribution on $\mathcal{D}$ w.r.t. a heldout validation set $\mathcal{D}'$. Adapting the gradient influence metric from Pruthi et al. (2020), we measure the influence of the gradients from training set $\mathcal{D}$ on the validation set $\mathcal{D}'$. Mathematically, this is defined as

$$\sum_{(\mathbf{x},\mathbf{y})\in\mathcal{D}} \sum_{(\mathbf{x}',\mathbf{y}')\in\mathcal{D}'} \frac{\cos(\boldsymbol{h}(\mathbf{x},\mathbf{y}), \boldsymbol{h}(\mathbf{x}',\mathbf{y}'))}{|\mathcal{D}| \cdot |\mathcal{D}'|}$$

Following Xia et al. (2024), we can additionally improve this definition by taking Adam optimization into account. This requires an initial warmup training of the model on an independent copy of $\mathcal{D}$; more details are provided in Section C.1.1.

We compare teachers for LLaMA-1B trained on GSM8K, with $\mathcal{D}'$ being the validation split consisting of human-written solutions. We also report the loss on the reference set $\mathcal{D}'$ of the trained models after the warmup phase following Xia et al. (2024). As shown in Figure 7, influence-based metrics show low correlation with the student performance after training.

### C.1.1  DETAILS ON LESS (XIA ET AL., 2024)

For discussion in this section, we will use a few additional notations: we will use $\Theta_{\mathcal{S}} \in \mathbb{R}^D$ to generally denote the parameters of the student model. $\Theta_{\mathcal{S}}^{(t)}$ will denote the parameters of the model at time $t$.

Suppose, we train our model with Adam (Kingma, 2014) with hyperparameters $\eta, \beta_1, \beta_2, \epsilon$, that tracks two gradient statistics: $\mathbf{m}_t$ and $\boldsymbol{v}_t$ (initialized at 0s) at each step of training. The gradient

update step at any step $t > 0$ with a gradient vector $\mathbf{g} \in \mathbb{R}^D$ is given by

$$\Theta_{\mathcal{S}}^{(t+1)} \leftarrow \Theta_{\mathcal{S}}^{(t)} - \eta_t \Gamma_{t+1}(\mathbf{g}),$$

$$\text{where} \quad \Gamma_{t+1}(\mathbf{g}) = \mathbf{m}_{t+1} \odot \frac{1}{\sqrt{\mathbf{v}_{t+1}} + \epsilon},$$

$$\mathbf{m}_{t+1} \leftarrow \frac{1}{1 - \beta_1^{t+1}} \left( (1 - \beta_1)\mathbf{m}_t + \beta_1 \mathbf{g} \right),$$

$$\boldsymbol{v}_{t+1} \leftarrow \frac{1}{1 - \beta_2^{t+1}} \left( (1 - \beta_2)\boldsymbol{v}_t + \beta_2 \left( \mathbf{g} \odot \mathbf{g} \right) \right),$$

where $\eta_t$ denotes the learning rate at time step $t$, that can be a function of $\eta$ and time-step $t$ depending on the learning rate schedule. Here $\Gamma$ denotes gradient pre-conditioning function that uses the gradient statistics $\mathbf{m}$ and $\boldsymbol{v}$.

LESS (Xia et al., 2024) considers the gradient update under Adam to compute the influence of the gradients from training dataset $\mathcal{D}$ on validation set $\mathcal{D}'$. Suppose in addition to $\mathcal{D}$, we have a warmup dataset $\mathcal{D}_{warmup}$. Then, the model is trained for $T$ steps (4 epochs) on the warmup dataset $\mathcal{D}_{warmup}$.

Let $\Gamma_{T/4}, \Gamma_{T/2}, \Gamma_{T/4}$, and $\Gamma_T$ be the pre-conditioning functions at step $T/4$, $T/2$, $3T/4$, and $T$ respectively. Furthermore, let $\bar{\eta}_{T/4}$ denote the average learning rate between timestep $0$ and $T/4$ (similarly, define $\bar{\eta}_{T/2}$ as the average learning rate between timestep $T/4$ and $T/2$, and so on). Then, the influence of each input $(\mathbf{x}, \mathbf{y}) \in \mathcal{D}$, denoted by $\mathbf{Inf}(\mathbf{x}, \mathbf{y})$, is given by

$$\mathbf{Inf}(\mathbf{x}, \mathbf{y}) := \frac{1}{4|\mathcal{D}'|} \sum_{(\mathbf{x}', \mathbf{y}') \in \mathcal{D}'} \sum_{i=1}^{4} \bar{\eta}_{iT/4} \text{cosine} \left( \Gamma_{iT/4}(\boldsymbol{g}(\mathbf{x}, \mathbf{y})), \boldsymbol{g}(\mathbf{x}', \mathbf{y}') \right). \tag{6}$$

For computational efficiency, we use a random projection $\Pi$:

$$\mathbf{Inf}(\mathbf{x}, \mathbf{y}) := \frac{1}{4|\mathcal{D}'|} \sum_{(\mathbf{x}', \mathbf{y}') \in \mathcal{D}'} \sum_{i=1}^{4} \bar{\eta}_{iT/4} \text{cosine} \left( \Pi\Gamma_{iT/4}(\boldsymbol{g}(\mathbf{x}, \mathbf{y})), \Pi\boldsymbol{g}(\mathbf{x}', \mathbf{y}') \right).$$

In LESS, $\mathbf{Inf}(\mathbf{x}, \mathbf{y})$ was used to pick the most influential data points in $\mathcal{D}$. We report the performance of LESS for teacher selection by using an average influence function on the dataset $\mathcal{D}$ to measure its overall quality:

$$\mathbf{Inf}(\mathcal{D}) = \frac{1}{|\mathcal{D}|} \sum_{(\mathbf{x}, \mathbf{y}) \in \mathcal{D}} \mathbf{Inf}(\mathbf{x}, \mathbf{y}).$$

### C.2 ISSUES WITH INFLUENCE BASED METRICS FOR TEACHER SELECTION

We hypothesize that a key limitation of these influence-based metrics is the choice of $\mathcal{D}'$: it should both reflect a common ground truth and effectively differentiate models without bias toward any particular one, which can be challenging to construct.

Gradient influence metrics like LESS approximate the decrease in validation loss when taking a gradient optimization step. On an update $\Theta_{\mathcal{S}}^{(t+1)}$ from $\Theta_{\mathcal{S}}^{(t)}$ using a gradient vector $\boldsymbol{g}$, with the corresponding student given by $\pi_{\mathcal{S}}^{(t+1)}$ and $\pi_{\mathcal{S}}^{(t)}$ respectively, we can show with a first order taylor expansion:

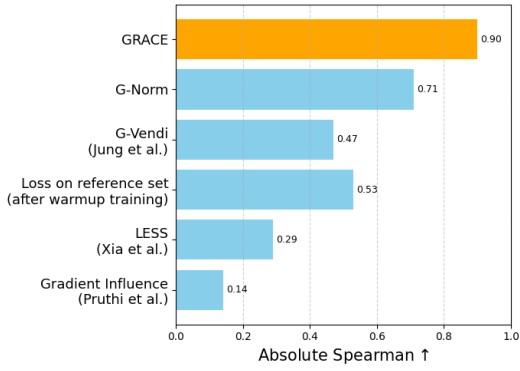

Figure 7: Influence-based measures that rely on a validation set are ineffective for selecting good teachers. We report LLaMA-1B performance on GSM8K across different teacher choices at a generation temperature of $0.6$. The validation split from Cobbe et al. (2021) serves as the held-out set for computing Gradient Influence and LESS (Pruthi et al., 2020; Xia et al., 2024).

$$\mathbb{E}_{(\mathbf{x}',\mathbf{y}')\in\mathcal{D}'}\left(\mathcal{L}(\mathbf{y}'|\mathbf{x}';\pi_{\mathcal{S}}^{(t+1)}) - \mathcal{L}(\mathbf{y}'|\mathbf{x}';\pi_{\mathcal{S}}^{(t)})\right) \propto \mathbb{E}_{(\mathbf{x}',\mathbf{y}')\in\mathcal{D}'}\left\langle \Theta_{\mathcal{S}}^{(t+1)} - \Theta_{\mathcal{S}}^{(t)}, \boldsymbol{g}(\mathbf{y}',\mathbf{x}')\right\rangle$$
$$= \mathbb{E}_{(\mathbf{x}',\mathbf{y}')\in\mathcal{D}'}\eta_t\left\langle \Gamma_{t+1}(\mathbf{g}), \boldsymbol{g}(\mathbf{y}',\mathbf{x}')\right\rangle. \qquad (7)$$

However, there are two primary components in the solution $\mathbf{y}'$: chain-of-thought and the final answer. If we denote each response $\mathbf{y}'$ as composition of $(\mathrm{cot}', a')$, then the above formulation can be re-formulated as

$$\mathbb{E}_{(\mathbf{x}',\mathbf{y}')\in\mathcal{D}':\mathbf{y}'=(\mathrm{cot}',a')}\left(\mathcal{L}(\mathrm{cot}',a'|\mathbf{x}';\pi_{\mathcal{S}}^{(t+1)}) - \mathcal{L}(\mathrm{cot}',a'|\mathbf{x}';\pi_{\mathcal{S}}^{(t)})\right).$$

For a given prompt $\mathbf{x}'$, the final answer will be the same across different responses but the chain-of-thought $\mathrm{cot}'$ can differ. For a pool of teachers $\{\mathcal{T}_1, \cdots, \mathcal{T}_p\}$ with training generations $\mathcal{D}_{\mathcal{T}_i}$ for each teacher, the chain-of-thought patterns differ across the teachers. Therefore, the validation set $\mathcal{D}'$ should ideally be constructed such that the student's loss or gradient metrics evaluated on it can effectively distinguish between the diverse chain-of-thought behaviors across all the teachers' training generations.

In Figure 7, we show that using the validation set from the GSM8k dataset, which has human-written answers to each question, as $\mathcal{D}'$ doesn't help effective teacher selection. In fact, the loss on the set $\mathcal{D}'$ itself doesn't help to effectively distinguish between different teachers. We keep principled selection of a common validation set $\mathcal{D}'$ for selecting the right teacher to future work.

### C.2.1 CAN WE USE TEACHER GENERATED VALIDATION SETS FOR EACH TEACHER?

An alternative is to use separate validation sets for different teachers, each containing responses generated by a specific teacher on a shared set of questions. That is, on a given validation set of prompts $\mathcal{D}'$, we simply collect the response of a teacher $\mathcal{T}$ to each prompt $\mathbf{x}' \in \mathcal{D}'$ and create validation set $\mathcal{D}'_{\mathcal{T}}$. Then, for each teacher $\mathcal{T}$, we modify the LESS influence metric on an example $(\mathbf{x}, \mathbf{y}) \in \mathcal{D}_{\mathcal{T}}$ from Equation (6) as

$$\mathbf{Inf}(\mathbf{x},\mathbf{y}) := \frac{1}{4|\mathcal{D}'_{\mathcal{T}}|}\sum_{(\mathbf{x}',\mathbf{y}')\in\mathcal{D}'_{\mathcal{T}}}\sum_{i=1}^{4}\bar{\eta}_{iT/4}\mathrm{cosine}\big(\Pi\Gamma_{iT/4}(\boldsymbol{g}(\mathbf{x},\mathbf{y})), \Pi\boldsymbol{g}(\mathbf{x}',\mathbf{y}')\big),$$

where $\Pi$ denotes a random projection matrix.

We report the performance of gradient influence metrics under this setup in Figure 8. We find that the performance of the gradient influence metrics for teacher selection do not substantially change under this setup, compared to using a common validation set in Figure 7.

One primary concern is that this approach breaks comparability across students trained on different teachers, as gradient estimates from different validation sets cannot be used to compare across them. That is, if $(\mathcal{D}_{\mathcal{T}_1}, \mathcal{D}'_{\mathcal{T}_1})$ and $(\mathcal{D}_{\mathcal{T}_2}, \mathcal{D}'_{\mathcal{T}_2})$ represent the training and validation sets from two teachers, influence estimate scores with $(\mathcal{D}_{\mathcal{T}_1}, \mathcal{D}'_{\mathcal{T}_1})$ and $(\mathcal{D}_{\mathcal{T}_2}, \mathcal{D}'_{\mathcal{T}_2})$ can't be used to directly differentiate between the generations of the teachers $\mathcal{T}_1$ and $\mathcal{T}_2$.

### C.3 DESCRIPTIONS OF OTHER BASELINES

We consider the following additional baselines:

1. **Student Loss on the teacher's generations:** This is defined by the average cross entropy loss of the student model on the teacher's generations before training.

2. **G-Norm**: Defined in Equation (3), this measures the variance of the gradients w.r.t. the mean gradient.

3. **G-Vendi**: Defined in Equation (2), this measures the directional entropy in the gradients.

4. **Determinant**: This is defined by the determinant of the gradient matrix $\tilde{\boldsymbol{G}}(\mathcal{D})$ and serves as notion of diversity of the gradients.

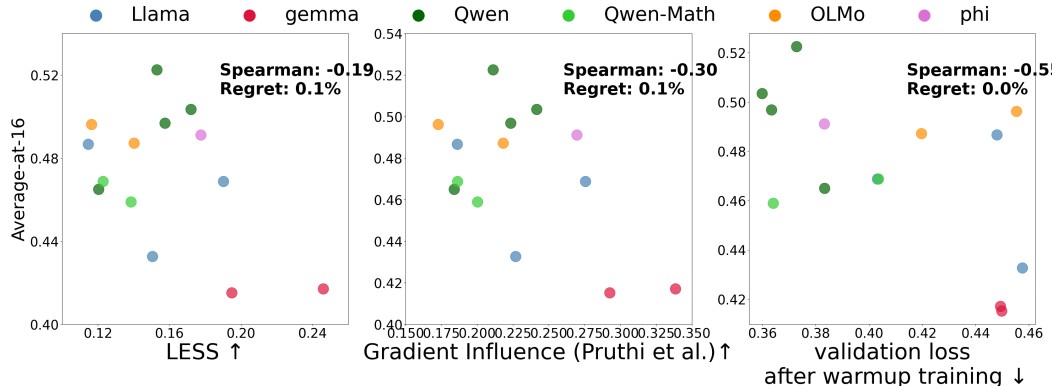

Figure 8: We repeat experiments from Figure 7. But instead of using a common validation set with human written answers from Cobbe et al. (2021) to measure the gradient influence metrics across the teachers, we use a separate validation set for each teacher, where we use the respective teacher's response on the validation set of questions. We then report the performance of LLaMA-1B on GSM8K across different teacher choices at a generation temperature of 0.6. We find that the teacher-selection performance of the influence-based metrics does not improve under this setup. For comparison, refer to Figure 21, where gradient-based metrics achieve notably higher Spearman correlation with student performance.

5. **Determinant $\times$ gradient norm**, corresponding to BADGE (Ash et al., 2019): This is defined by the determinant of the gradient matrix $\boldsymbol{G}(\mathcal{D})$ and serves as notion of diversity of the gradients, that also takes their norm into account.

6. **Gradient cosine**, which is another way to capture gradient diversity: Given gradients from the training set $\mathcal{D}$, we compute pairwise inner product between the normalized gradients of generations for the same prompt:

$$\mathbb{E}_{\mathbf{x}}\mathbb{E}_{(\mathbf{x},\mathbf{y}_1),(\mathbf{x},\mathbf{y}_2)\sim\mathcal{D}}\left[\frac{\mathbf{g_1}}{\|\mathbf{g}_1\|_2}\right]^{\top}\frac{\mathbf{g_2}}{\|\mathbf{g}_2\|_2},$$
$$\text{where } \mathbf{g}_1 = \mathbf{h}(\mathbf{x},\mathbf{y}_1),$$
$$\mathbf{g}_2 = \mathbf{h}(\mathbf{x},\mathbf{y}_2).$$

7. **Gradient inner product**: we compute pairwise inner product between the gradients of generations from the same prompt.

$$\mathbb{E}_{\mathbf{x}}\mathbb{E}_{(\mathbf{x},\mathbf{y}_1),(\mathbf{x},\mathbf{y}_2)\sim\mathcal{D}}\,\mathbf{g_1}^{\top}\mathbf{g_2},$$
$$\text{where } \mathbf{g}_1 = \mathbf{h}(\mathbf{x},\mathbf{y}_1),$$
$$\mathbf{g}_2 = \mathbf{h}(\mathbf{x},\mathbf{y}_2).$$

This can be considered as considering gradient magnitude in addition to the gradient cosine.

8. **Average Probabilities** (per token): this computes the average probability per token of the student on the teacher's generations, averaged over all generations and all prompts.

9. **Average length:** this computes the average length of the teacher's generations.

As mentioned in Section 3, naive metrics are not useful for identifying the best teachers.

# D    CONNECTING GRACE TO LEAVE-ONE-OUT CONDITIONAL MUTUAL INFORMATION (LOO-CMI)

## D.1    BACKGROUND ON LOO-CMI

We start with a general description of the LOO-CMI framework from Steinke & Zakynthinou (2020); Rammal et al. (2022). In this setup, we have a dataset $\mathcal{D} = \{Z_1, Z_2, \cdots, Z_n\}$ of $n$ i.i.d. samples

drawn from the parent distribution $\mathcal{P}$, where $Z_i$ refers to a general datapoint containing the data-label pair $(\mathbf{x}, \mathbf{y})$.

Consider a training algorithm $\mathcal{A} : \mathcal{D} \to \Theta$ that takes a training set $\mathcal{D}$ and returns a weight parameter $\Theta$. Let $u$ be an index uniformly drawn from $\{1, \cdots, n\}$, and denote by $\mathcal{D}_{-u} := \mathcal{D} \setminus \{Z_u\}$ the size-$(n-1)$ dataset with the $u$-th sample removed. Denote the output of the training algorithm on $\mathcal{D}_{-u}$ by $\Theta_{-u} = \mathcal{A}(\mathcal{D}_{-u})$. Then, the *leave-one-out conditional mutual information (LOO-CMI)* of $\mathcal{A}$ is defined as

$$I_{\text{LOO}}(\mathcal{A}; \mathcal{D}) := I(\Theta_{-U}; U \mid \mathcal{D}) = \mathbb{E}_{u \sim \text{Unif}[n]} \left[ \text{KL}(P_{\Theta_{-u}|u,\mathcal{D}} \| P_{\Theta_{-u}|\mathcal{D}}) \right] \tag{8}$$

where $I(\cdot; \cdot \mid \cdot)$ denotes the conditional mutual information, $\text{KL}(\cdot \| \cdot)$ the Kullback-Leibler divergence and $P_\Theta$ denotes the distribution over parameters $\Theta$, where the randomness comes from gradient noise. The LOO-CMI quantifies how much information the trained model parameters $\Theta$ reveal about which sample index $U$ was removed when training on the remaining data. A small value of $I_{\text{LOO}}(\Theta; \mathcal{D})$ indicates that the algorithm's output is insensitive to the removal of a single training example, which corresponds to higher algorithmic stability and better generalization.

**Generalization bound.**   With a slight abuse of notation, let $\ell : \Theta \times \mathcal{D} \to [0, 1]$ denote the loss on a dataset $\mathcal{D}$, and let $\ell : \Theta \to [0, 1]$ denote the population loss, i.e. taking expectation over $\mathcal{P}$. The generalization gap of a randomized algorithm $\mathcal{A}$ that has trained on a dataset $\mathcal{D}$ is measured as

$$\text{gen}(\mathcal{A}) := \left| \mathbb{E}_{\mathcal{A}, \mathcal{D} \sim \mathcal{P}, |\mathcal{D}| = n-1} \left[ \ell(\mathcal{A}(\mathcal{D})) - \ell(\mathcal{A}(\mathcal{D}), \mathcal{D}) \right] \right|.$$

Rammal et al. (2022) show that the expected generalization gap of $\mathcal{A}$ satisfies

$$\text{gen}(\mathcal{A}) \leq \frac{n}{\sqrt{2}(n-1)} \mathbb{E}_{\mathcal{D} \sim \mathcal{P}} \sqrt{I_{\text{LOO}}(\mathcal{A}; \mathcal{D})}. \tag{9}$$

### D.2   Connecting GRACE to LOO-CMI

Our discussions below focus on the original gradients without pre-processing.

**Notations.**   Let $\Theta_\mathcal{S} \in \mathbb{R}^D$ denote the parameters of the student model before training. For the theoretical analysis, we use $m = 1$ teacher response per prompt on the $n$ prompts in the training set $\mathcal{D}$, though our results can be generalized to $m > 1$. From now on, we will use the uniformly random prompt-generation pair $(\mathbf{x}, \mathbf{y}) \in \mathcal{D}$ in place of the scale random variable $u \sim \text{Unif}([n])$ where $|\mathcal{D}| = n$. For example, we will use $\mathcal{D}_{-(\mathbf{x}, \mathbf{y})}$ to denote the dataset with $(\mathbf{x}, \mathbf{y})$ removed from $\mathcal{D}$. For simplicity, we will drop $\Theta_\mathcal{S}$ from all our notations involving $\mathbf{M}$ and $\mathbf{g}$.

We will use $\hat{\mathbb{E}}$ to denote the empirical expectation over $\mathcal{D}$. For each $(\mathbf{x}, \mathbf{y})$, we perform a single gradient update with a preconditioner matrix $\mathbf{M}$ that can depend on the training set $\mathcal{D}_{-(\mathbf{x}, \mathbf{y})}$:

$$\Theta_{-(\mathbf{x}, \mathbf{y})} \leftarrow \Theta_\mathcal{S} - \eta \hat{\mathbb{E}}_{(\bar{\mathbf{x}}, \bar{\mathbf{y}}) \in \mathcal{D}_{-(\mathbf{x}, \mathbf{y})}} \left[ \mathbf{M}(\mathcal{D}_{-(\mathbf{x}, \mathbf{y})}) \mathbf{g}(\bar{\mathbf{x}}, \bar{\mathbf{y}}) \right] + \epsilon, \tag{10}$$

where $\epsilon \sim \mathcal{N}(0, \sigma^2 \mathbf{I})$ denotes a Gaussian noise, which represents the randomness in our training algorithm.

For any dataset $\hat{\mathcal{D}}$, we define the following notations:

$$\mu(\hat{\mathcal{D}}) = \hat{\mathbb{E}}_{(\mathbf{x}, \mathbf{y}) \in \hat{\mathcal{D}}}[\mathbf{g}(\mathbf{x}, \mathbf{y})], \quad \tilde{\Sigma}(\hat{\mathcal{D}}) = \frac{1}{|\hat{\mathcal{D}}|} \tilde{\mathbf{G}}(\hat{\mathcal{D}})^\top \tilde{\mathbf{G}}(\hat{\mathcal{D}}), \quad \hat{\Sigma}(\hat{\mathcal{D}}) = \tilde{\Sigma}(\hat{\mathcal{D}}) + \frac{\nu}{d} \boldsymbol{I},$$

where $\nu > 0$ is added for numerical stability.

We will further use the following shorthands:

$$\mu_{-(\mathbf{x}, \mathbf{y})} := \mu \left( \mathcal{D}_{-(\mathbf{x}, \mathbf{y})} \right), \quad \mathbf{M}_{-(\mathbf{x}, \mathbf{y})} := \mathbf{M} \left( \mathcal{D}_{-(\mathbf{x}, \mathbf{y})} \right).$$

Then, the CMI bounds for the preconditioned update step in Equation (10) are given by:

**Lemma 2** (Bounds for Pre-conditioned Gradient Descent). *Suppose* $\exists \beta, B > 0$ *such that* $\left\| \mathbf{M}_{-(\mathbf{x},\mathbf{y})} - \mathbf{M}_{-(\bar{\mathbf{x}},\bar{\mathbf{y}})} \right\|_2 \leq \frac{\beta}{n}$ *for all* $(\bar{\mathbf{x}}, \bar{\mathbf{y}}), (\mathbf{x}, \mathbf{y}) \in \mathcal{D}$, *and* $\|\mathbf{g}(\mathbf{x}, \mathbf{y})\| \leq B$ *for all* $(\mathbf{x}, \mathbf{y}) \in \mathcal{D}$. *Under the one-step update rule on the parameter* $\Theta$ *with learning rate* $\eta$ *(Equation* (10)*) as algorithm* $\mathcal{A}$, *the LOO-CMI can be bounded as*

$$I_{\text{LOO}}(\mathcal{A}; \mathcal{D}) \leq \frac{3\eta^2}{\sigma^2(n-1)^2} \quad \hat{\mathbb{E}}_{(\mathbf{x},\mathbf{y}) \in \mathcal{D}} \quad \left\| \mathbf{M}_{-(\mathbf{x},\mathbf{y})} \mathbf{g}_\mu(\mathbf{x}, \mathbf{y}) \right\|_2^2$$

$$+ \frac{3\eta^2}{\sigma^2} \hat{\mathbb{E}}_{(\mathbf{x},\mathbf{y}) \in \mathcal{D}} \left\| \left( \mathbf{M}_{-(\mathbf{x},\mathbf{y})} - \hat{\mathbb{E}}_{(\hat{\mathbf{x}},\hat{\mathbf{y}}) \in \mathcal{D}} \mathbf{M}_{-(\hat{\mathbf{x}},\hat{\mathbf{y}})} \right) \mu(\mathcal{D}) \right\|_2^2 + \mathcal{O}(\beta/n^4), \quad (11)$$

*where* $\mathbf{g}_\mu(\mathbf{x}, \mathbf{y}) = \mathbf{g}(\mathbf{x}, \mathbf{y}) - \mu(\mathcal{D})$.

**Special cases:** There are two notable special cases: the first is when $\mathbf{M} = \mathbf{I}$, corresponding to update step in the standard gradient descent update algorithm. The second is when $\mathbf{M} = \hat{\Sigma}^{-1/2}$, which denotes the preconditioner matrix used to define GRACE.

**Corollary D.1** (Connecting CMI to G-Norm). *When* $\mathbf{M}$ *is set as identity matrix* $\mathbf{I}$ *for all* $(\mathbf{x}, \mathbf{y}) \in \mathcal{D}$, *then LOO-CMI for the one-step update rule on the parameter* $\Theta$ *with learning rate* $\eta$ *given by*

$$I_{\text{LOO}}(\mathcal{A}; \mathcal{D}) \leq \frac{3\eta^2}{\sigma^2(n-1)^2} \hat{\mathbb{E}}_{(\mathbf{x},\mathbf{y}) \in \mathcal{D}} \quad \left\| \mathbf{g}_\mu(\mathbf{x}, \mathbf{y}) \right\|_2^2$$

*where* $\mathbf{g}_\mu(\mathbf{x}, \mathbf{y}) = \mathbf{g}(\mathbf{x}, \mathbf{y}) - \mu(\mathcal{D})$

*This connects CMI to the G-Norm (Equation* (3)*) metric.*

**Corollary D.2** (Connecting CMI to GRACE). $\mathbf{M}(\hat{\mathcal{D}})$ *is set as* $\hat{\Sigma}(\hat{\mathcal{D}})^{-1/2}$ *for any training set* $\hat{\mathcal{D}}$. *Denote* $\hat{\Sigma}_{-\{(\mathbf{x},\mathbf{y})\}} := \hat{\Sigma}(\mathcal{D}_{-\{(\mathbf{x},\mathbf{y})\}})$ *for any* $(\mathbf{x}, \mathbf{y})$ *pair for simplicity. Suppose there exists a constant* $0 < \alpha < 1$, *s.t. for any* $(\mathbf{x}, \mathbf{y}) \in \mathcal{D}$,

$$\left\| \left( \hat{\Sigma}_{-(\mathbf{x},\mathbf{y})}^{-1/2} - \hat{\mathbb{E}}_{(\hat{\mathbf{x}},\hat{\mathbf{y}}) \in \mathcal{D}} \hat{\Sigma}_{-(\hat{\mathbf{x}},\hat{\mathbf{y}})}^{-1/2} \right) \mu(\mathcal{D}) \right\|_2^2 < \frac{\alpha}{n^2} \left\| \hat{\Sigma}_{-(\mathbf{x},\mathbf{y})}^{-1/2} \mathbf{g}_\mu(\mathbf{x}, \mathbf{y}) \right\|_2^2. \quad (12)$$

*Then LOO-CMI for the one-step update rule on the parameter* $\Theta$ *with learning rate* $\eta$ *is bounded as*

$$I_{\text{LOO}}(\mathcal{A}; \mathcal{D}) \leq \frac{3(1+\alpha)\eta^2}{\sigma^2(n-1)^2} \quad \hat{\mathbb{E}}_{(\mathbf{x},\mathbf{y}) \in \mathcal{D}} \quad \left\| \hat{\Sigma}_{-(\mathbf{x},\mathbf{y})}^{-1/2} \mathbf{g}_\mu(\mathbf{x}, \mathbf{y}) \right\|_2^2 + \mathcal{O}(\beta/n^4)$$

$$:= \frac{3\eta^2(1+\alpha)}{\sigma^2(n-1)^2} \text{GRACE-Variance}(\mathcal{D}) + \mathcal{O}(\beta/n^4)$$

$$\leq \frac{3\eta^2(1+\alpha)}{\sigma^2(n-1)^2} \text{GRACE}(\mathcal{D}) + \mathcal{O}(\beta/n^4)$$

*where* $\mathbf{g}_\mu(\mathbf{x}, \mathbf{y}) = \mathbf{g}(\mathbf{x}, \mathbf{y}) - \mu(\mathcal{D})$.

*This connects CMI to the GRACE (Equation* (4)*) score with* $n$*-fold cross-validation.*

Theorem D.2 indicates that GRACE evaluates the stability of a one-step gradient update when a few prompts are removed from the batch. Importantly, the outcome of this update depends on the optimization method, since gradient descent and preconditioned updates can behave differently. In our setting, the preconditioner matrix is closely related to the one used in AdaGrad (Duchi et al., 2011). Since adaptive optimizers are the de facto choice for training language models, it is essential to incorporate this preconditioning effect in our analysis.

However, the pre-conditioner matrix that we use doesn't exactly match the optimization states of the Adam optimization algorithm that we use in practice, leaving space for further improving the definition of GRACE. This might require a short warm-up training phase of the student model and setting $\mathbf{M}$ as a function of the optimizer states during the warm-up training, akin to Xia et al. (2024). We leave a more thorough exploration of this direction to future work.

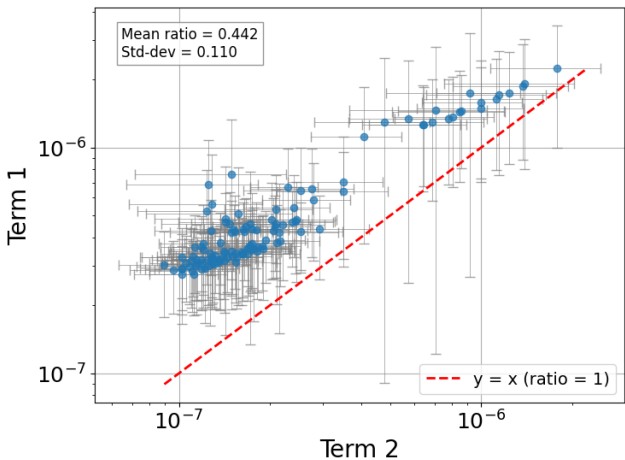

Figure 9: Validating assumption Equation (12) in Theorem D.2 using teachers on LLaMA-1B and GSM8K using $n = 512$. Here Term 1 is given by $\frac{1}{n^2} \left\| \hat{\Sigma}_{-(\mathbf{x},\mathbf{y})}^{-1/2} \mathbf{g}_\mu(\mathbf{x},\mathbf{y}) \right\|_2$, Term 2 is given by $\left\| \left( \hat{\Sigma}_{-(\mathbf{x},\mathbf{y})}^{-1/2} - \hat{\mathbb{E}}_{(\hat{\mathbf{x}},\hat{\mathbf{y}})\in\mathcal{D}} \hat{\Sigma}_{-(\hat{\mathbf{x}},\hat{\mathbf{y}})}^{-1/2} \right) \mu(\mathcal{D}) \right\|_2^2$. $\alpha$ is defined as Term 2 / Term 1. **Main takeaway:** Term 1 dominates Term 2 across all teachers, with mean $\alpha = 0.442$, validating the assumption in Theorem D.2.

**Note on theoretical limitations:** Our current analysis establishes a connection between GRACE and an upper bound on the LOO-CMI, which has been shown to upper bound the generalization gap in terms of the loss (Rammal et al., 2022). However, Lemma 2 is stated only for a single gradient step, while in practice we compare the student's performance across different teachers after extensive training. Moreover, the loss may not tightly reflect the performance metric of interest. In particular, the student model's test loss after training does not correlate with the student performance when compared across different teachers, as shown by our empirical results. [5] Such theory-practice gap highlights the need for further theoretical understanding of GRACE, which we leave to future work.

**Validating assumption in Equation (12):** For simplicity, denote the two terms of interest as

$$\text{Term 1:} \frac{1}{n^2} \left\| \hat{\Sigma}_{-(\mathbf{x},\mathbf{y})}^{-1/2} \mathbf{g}_\mu(\mathbf{x},\mathbf{y}) \right\|_2$$

$$\text{Term 2:} \left\| \left( \hat{\Sigma}_{-(\mathbf{x},\mathbf{y})}^{-1/2} - \hat{\mathbb{E}}_{(\hat{\mathbf{x}},\hat{\mathbf{y}})\in\mathcal{D}} \hat{\Sigma}_{-(\hat{\mathbf{x}},\hat{\mathbf{y}})}^{-1/2} \right) \mu(\mathcal{D}) \right\|_2^2.$$

Expectation of Term 1 gives GRACE with $C = n$ and is the main term of interest. Term 2 measures the stability of the pre-conditioning matrix $\tilde{\Sigma}$ when a prompt is dropped. The ratio of Term 1 and Term 2 measures how dominant is Term 1 compared to Term 2 in Equation (11), formally referred to as $\alpha$ in Equation (12). If $\alpha < 1$, this implies the bound on the conditional mutual information is primarily driven by our formulation of GRACE. Empirically, on LLaMA-1B training on GSM8K, we show that $\alpha$ has an average value between $0.442$ across all teachers in Figure 9.

### D.2.1 PROOF OF LEMMA 2

*Proof.* For any $(\mathbf{x}, \mathbf{y})$ pair, denote the mean parameter update on the training set $\mathcal{D}_{-(\mathbf{x},\mathbf{y})}$ as $\delta_{-(\mathbf{x},\mathbf{y})} := -\eta \mathbf{M}_{-(\mathbf{x},\mathbf{y})} \mu_{-(\mathbf{x},\mathbf{y})}$. Starting from student parameters $\Theta_\mathcal{S}$, the update rule for any set $\mathcal{D} \setminus \{(\mathbf{x}, \mathbf{y})\}$ is given by

$$\Theta_{-(\mathbf{x},\mathbf{y})} \leftarrow \Theta_\mathcal{S} + \delta_{-(\mathbf{x},\mathbf{y})} + \epsilon,$$

where $\epsilon \sim \mathcal{N}(0, \sigma^2 I)$ is a Gaussian noise and captures the randomness in the algorithm. Hence $\Theta_{-(\mathbf{x},\mathbf{y})}$ follows the distribution

$$\Theta_{-(\mathbf{x},\mathbf{y})} \sim \mathcal{N}\left( \Theta_\mathcal{S} + \delta_{-(\mathbf{x},\mathbf{y})}, \sigma^2 \mathbf{I} \right).$$

---

[5]We will extensively discuss the comparison of GRACE and loss in Section D.3.

Then, using the properties of Gaussian distribution;

$$I_{\text{LOO}}(\mathcal{A}; \mathcal{D}) = \hat{\mathbb{E}}_{(\mathbf{x},\mathbf{y}) \in \mathcal{D}} D_{\text{KL}}\left(P_{\Theta_{-(\mathbf{x},\mathbf{y})}} \middle\| \hat{\mathbb{E}}_{(\hat{\mathbf{x}},\hat{\mathbf{y}}) \in \mathcal{D}} P_{\Theta_{-(\hat{\mathbf{x}},\hat{\mathbf{y}})}}\right)$$

$$= \hat{\mathbb{E}}_{(\mathbf{x},\mathbf{y}) \in \mathcal{D}} \mathbb{E}_{\zeta \sim \mathcal{N}\left(\delta_{-(\mathbf{x},\mathbf{y})}, \sigma^2 \mathbf{I}\right)} \left(\log\left(\tfrac{1}{Z} e^{-\|\zeta - \delta_{-(\mathbf{x},\mathbf{y})}\|_2^2 / 2\sigma^2}\right) - \log \hat{\mathbb{E}}_{(\hat{\mathbf{x}},\hat{\mathbf{y}}) \in \mathcal{D}} \left(\tfrac{1}{Z} e^{-\|\zeta - \delta_{-(\hat{\mathbf{x}},\hat{\mathbf{y}})}\|_2^2 / 2\sigma^2}\right)\right) \tag{13}$$

$$\leq \hat{\mathbb{E}}_{(\mathbf{x},\mathbf{y}) \in \mathcal{D}} \mathbb{E}_{\zeta \sim \mathcal{N}\left(\delta_{-(\mathbf{x},\mathbf{y})}, \sigma^2 \mathbf{I}\right)} \left(\log\left(\tfrac{1}{Z} e^{-\|\zeta - \delta_{-(\mathbf{x},\mathbf{y})}\|_2^2 / 2\sigma^2}\right) - \hat{\mathbb{E}}_{(\hat{\mathbf{x}},\hat{\mathbf{y}}) \in \mathcal{D}} \log\left(\tfrac{1}{Z} e^{-\|\zeta - \delta_{-(\hat{\mathbf{x}},\hat{\mathbf{y}})}\|_2^2 / 2\sigma^2}\right)\right) \tag{14}$$

$$= \frac{1}{2\sigma^2} \hat{\mathbb{E}}_{(\mathbf{x},\mathbf{y}) \in \mathcal{D}} \mathbb{E}_{\zeta \sim \mathcal{N}\left(\delta_{-(\mathbf{x},\mathbf{y})}, \sigma^2 \mathbf{I}\right)} \left(-\|\zeta - \delta_{-(\mathbf{x},\mathbf{y})}\|_2^2 + \hat{\mathbb{E}}_{(\hat{\mathbf{x}},\hat{\mathbf{y}}) \in \mathcal{D}} \|\zeta - \delta_{-(\hat{\mathbf{x}},\hat{\mathbf{y}})}\|_2^2\right)$$

$$= \frac{1}{2\sigma^2} \hat{\mathbb{E}}_{(\mathbf{x},\mathbf{y}) \in \mathcal{D}} \hat{\mathbb{E}}_{(\hat{\mathbf{x}},\hat{\mathbf{y}}) \in \mathcal{D}} \mathbb{E}_{\zeta \sim \mathcal{N}\left(\delta_{-(\mathbf{x},\mathbf{y})}, \sigma^2 \mathbf{I}\right)} \left(-\|\zeta - \delta_{-(\mathbf{x},\mathbf{y})}\|_2^2 + \|\zeta - \delta_{-(\hat{\mathbf{x}},\hat{\mathbf{y}})}\|_2^2\right)$$

$$= \frac{1}{2\sigma^2} \hat{\mathbb{E}}_{(\mathbf{x},\mathbf{y}) \in \mathcal{D}} \hat{\mathbb{E}}_{(\hat{\mathbf{x}},\hat{\mathbf{y}}) \in \mathcal{D}} \left\|\delta_{-(\hat{\mathbf{x}},\hat{\mathbf{y}})} - \delta_{-(\mathbf{x},\mathbf{y})}\right\|_2^2 \tag{15}$$

$$= \frac{1}{\sigma^2} \hat{\mathbb{E}}_{(\mathbf{x},\mathbf{y}) \in \mathcal{D}} \left\|\delta_{-(\mathbf{x},\mathbf{y})} - \hat{\mathbb{E}}_{(\hat{\mathbf{x}},\hat{\mathbf{y}}) \in \mathcal{D}} \delta_{-(\hat{\mathbf{x}},\hat{\mathbf{y}})}\right\|_2^2 \tag{16}$$

In the second step (Equation (13)), we simply use the CDF formulation of gaussian distribution, where $Z = (2\pi e)^{-D}$. The third step applies a jensen's inequality (Equation (14)). In Equation (15), we use for any two vectors $\mathbf{a}, \mathbf{b}$:

$$\mathbb{E}_{\zeta \sim \mathcal{N}(\mathbf{a}, \sigma^2 \mathbf{I})} \left(-\|\zeta - \mathbf{a}\|_2^2 + \|\zeta - \mathbf{b}\|_2^2\right) = \mathbb{E}_{\zeta \sim \mathcal{N}(\mathbf{a}, \sigma^2 \mathbf{I})} 2(\mathbf{a} - \mathbf{b})^\top \zeta - \|\mathbf{a}\|_2^2 + \|\mathbf{b}\|_2^2$$

$$= 2(\mathbf{a} - \mathbf{b})^\top \mathbf{a} - \|\mathbf{a}\|_2^2 + \|\mathbf{b}\|_2^2$$

$$= \|\mathbf{b} - \mathbf{a}\|_2^2.$$

Finally, Equation (16) follows because both $\delta_{-(\mathbf{x},\mathbf{y})}$ and $\delta_{-(\hat{\mathbf{x}},\hat{\mathbf{y}})}$ come from the identical set.

Using the definition of $\delta$, we have from Equation (16):

$$\mathbb{E}_{u \sim \text{Unif}([n])} I(\Theta_{-u}; u \mid \mathcal{D}) \leq \frac{1}{\sigma^2} \hat{\mathbb{E}}_{(\mathbf{x},\mathbf{y}) \in \mathcal{D}} \left\|\mathbf{M}_{-(\mathbf{x},\mathbf{y})} \mu_{-(\mathbf{x},\mathbf{y})} - \hat{\mathbb{E}}_{(\hat{\mathbf{x}},\hat{\mathbf{y}}) \in \mathcal{D}} \mathbf{M}_{-(\hat{\mathbf{x}},\hat{\mathbf{y}})} \mu_{-(\hat{\mathbf{x}},\hat{\mathbf{y}})}\right\|_2^2$$

**Warmup: When the pre-conditioner is identity matrix** Then for any $(\mathbf{x}, \mathbf{y})$ pair, we have $\mathbf{M}_{-(\mathbf{x},\mathbf{y})} = \mathbf{I}$. Then, the formulation simplifies to

$$\mathbb{E}_{u \sim \text{Unif}([n])} I(\Theta_{-u}; u \mid \mathcal{D})$$

$$\leq \frac{\eta^2}{\sigma^2} \hat{\mathbb{E}}_{(\mathbf{x},\mathbf{y}) \in \mathcal{D}} \left\|\mu_{-(\mathbf{x},\mathbf{y})} - \hat{\mathbb{E}}_{(\hat{\mathbf{x}},\hat{\mathbf{y}}) \in \mathcal{D}} \mu_{-(\hat{\mathbf{x}},\hat{\mathbf{y}})}\right\|_2^2$$

$$= \frac{\eta^2}{\sigma^2} \hat{\mathbb{E}}_{(\mathbf{x},\mathbf{y}) \in \mathcal{D}} \left\|\frac{n}{n-1}\mu(\mathcal{D}) - \frac{1}{n-1}\mathbf{g}(\mathbf{x}, \mathbf{y}) - \hat{\mathbb{E}}_{(\hat{\mathbf{x}},\hat{\mathbf{y}}) \in \mathcal{D}} \left(\frac{n}{n-1}\mu(\mathcal{D}) - \frac{1}{n-1}\mathbf{g}(\hat{\mathbf{x}}, \hat{\mathbf{y}})\right)\right\|_2^2 \tag{17}$$

$$= \frac{\eta^2}{\sigma^2(n-1)^2} \hat{\mathbb{E}}_{(\mathbf{x},\mathbf{y}) \in \mathcal{D}} \left\|\mathbf{g}(\mathbf{x}, \mathbf{y}) - \hat{\mathbb{E}}_{(\hat{\mathbf{x}},\hat{\mathbf{y}}) \in \mathcal{D}} \mathbf{g}(\hat{\mathbf{x}}, \hat{\mathbf{y}})\right\|_2^2$$

$$:= \frac{\eta^2}{\sigma^2(n-1)^2} \hat{\mathbb{E}}_{(\mathbf{x},\mathbf{y}) \in \mathcal{D}} \|\mathbf{g}(\mathbf{x}, \mathbf{y}) - \mu(\mathcal{D})\|_2^2 := \frac{\eta^2}{\sigma^2(n-1)^2} \hat{\mathbb{E}}_{(\mathbf{x},\mathbf{y}) \in \mathcal{D}} \|\mathbf{g}_\mu(\mathbf{x}, \mathbf{y})\|_2^2$$

The first step (Equation (17)) follows from the fact that $\mu(\mathcal{D}) = \hat{\mathbb{E}}_{(\hat{\mathbf{x}},\hat{\mathbf{y}}) \in \mathcal{D}} \mathbf{g}(\hat{\mathbf{x}}, \hat{\mathbf{y}}) := \frac{1}{n} \sum_{(\hat{\mathbf{x}},\hat{\mathbf{y}}) \in \mathcal{D}} \mathbf{g}(\hat{\mathbf{x}}, \hat{\mathbf{y}})$ and $\mu_{-(\mathbf{x},\mathbf{y})} = \frac{1}{n-1} \sum_{(\mathbf{x}',\mathbf{y}') \in \mathcal{D}_{-(\mathbf{x},\mathbf{y})}} \mathbf{g}(\mathbf{x}', \mathbf{y}')$.

**General pre-conditioner M:** We follow similar steps as above:

$$\mathbb{E}_{u \sim \text{Unif}([n])} I(\Theta_{-u}; u \mid \mathcal{D})$$

$$\leq \frac{\eta^2}{\sigma^2} \hat{\mathbb{E}}_{(\mathbf{x},\mathbf{y}) \in \mathcal{D}} \left\|\mathbf{M}_{-(\mathbf{x},\mathbf{y})} \mu_{-(\mathbf{x},\mathbf{y})} - \hat{\mathbb{E}}_{(\hat{\mathbf{x}},\hat{\mathbf{y}}) \in \mathcal{D}} \mathbf{M}_{-(\hat{\mathbf{x}},\hat{\mathbf{y}})} \mu_{-(\hat{\mathbf{x}},\hat{\mathbf{y}})}\right\|_2^2$$

$$= \frac{\eta^2}{\sigma^2} \hat{\mathbb{E}}_{(\mathbf{x},\mathbf{y}) \in \mathcal{D}} \Big\| \frac{n}{n-1} \mathbf{M}_{-(\mathbf{x},\mathbf{y})} \mu(\mathcal{D}) - \frac{1}{n-1} \mathbf{M}_{-(\mathbf{x},\mathbf{y})} \mathbf{g}(\mathbf{x},\mathbf{y})$$
$$- \hat{\mathbb{E}}_{(\hat{\mathbf{x}},\hat{\mathbf{y}}) \in \mathcal{D}} \left( \frac{n}{n-1} \mathbf{M}_{-(\hat{\mathbf{x}},\hat{\mathbf{y}})} \mu(\mathcal{D}) - \frac{1}{n-1} \mathbf{M}_{-(\hat{\mathbf{x}},\hat{\mathbf{y}})} \mathbf{g}(\hat{\mathbf{x}},\hat{\mathbf{y}}) \right) \Big\|^2$$
$$\text{(similar to Equation (17))}$$

$$= \frac{\eta^2}{\sigma^2} \hat{\mathbb{E}}_{(\mathbf{x},\mathbf{y}) \in \mathcal{D}} \Big\| \frac{n}{n-1} \left( \mathbf{M}_{-(\mathbf{x},\mathbf{y})} - \hat{\mathbb{E}}_{(\hat{\mathbf{x}},\hat{\mathbf{y}}) \in \mathcal{D}} \mathbf{M}_{-(\hat{\mathbf{x}},\hat{\mathbf{y}})} \right) \mu(\mathcal{D})$$
$$- \frac{1}{n-1} \left( \mathbf{M}_{-(\mathbf{x},\mathbf{y})} \mathbf{g}(\mathbf{x},\mathbf{y}) - \hat{\mathbb{E}}_{(\hat{\mathbf{x}},\hat{\mathbf{y}}) \in \mathcal{D}} \left( \mathbf{M}_{-(\hat{\mathbf{x}},\hat{\mathbf{y}})} \mathbf{g}(\hat{\mathbf{x}},\hat{\mathbf{y}}) \right) \right) \Big\|^2$$

$$= \frac{\eta^2}{\sigma^2} \hat{\mathbb{E}}_{(\mathbf{x},\mathbf{y}) \in \mathcal{D}} \Big\| \frac{n}{n-1} \left( \mathbf{M}_{-(\mathbf{x},\mathbf{y})} - \hat{\mathbb{E}}_{(\hat{\mathbf{x}},\hat{\mathbf{y}}) \in \mathcal{D}} \mathbf{M}_{-(\hat{\mathbf{x}},\hat{\mathbf{y}})} \right) \mu(\mathcal{D})$$
$$- \frac{1}{n-1} \mathbf{M}_{-(\mathbf{x},\mathbf{y})} \left( \mathbf{g}(\mathbf{x},\mathbf{y}) - \hat{\mathbb{E}}_{(\hat{\mathbf{x}},\hat{\mathbf{y}}) \in \mathcal{D}} \mathbf{g}(\hat{\mathbf{x}},\hat{\mathbf{y}}) \right)$$
$$- \frac{1}{n-1} \hat{\mathbb{E}}_{(\hat{\mathbf{x}},\hat{\mathbf{y}}) \in \mathcal{D}} \left( \mathbf{M}_{-(\mathbf{x},\mathbf{y})} - \mathbf{M}_{-(\hat{\mathbf{x}},\hat{\mathbf{y}})} \right) \mathbf{g}(\hat{\mathbf{x}},\hat{\mathbf{y}}) \Big\|^2$$

$$\leq \frac{3\eta^2}{\sigma^2} \hat{\mathbb{E}}_{(\mathbf{x},\mathbf{y}) \in \mathcal{D}} \left( \frac{n}{n-1} \right)^2 \hat{\mathbb{E}}_{(\mathbf{x},\mathbf{y}) \in \mathcal{D}} \left\| \left( \mathbf{M}_{-(\mathbf{x},\mathbf{y})} - \hat{\mathbb{E}}_{(\hat{\mathbf{x}},\hat{\mathbf{y}}) \in \mathcal{D}} \mathbf{M}_{-(\hat{\mathbf{x}},\hat{\mathbf{y}})} \right) \mu(\mathcal{D}) \right\|_2^2$$
$$+ \frac{3\eta^2}{\sigma^2} \frac{1}{(n-1)^2} \hat{\mathbb{E}}_{(\mathbf{x},\mathbf{y}) \in \mathcal{D}} \left\| \hat{\mathbb{E}}_{(\hat{\mathbf{x}},\hat{\mathbf{y}}) \in \mathcal{D}} \left( \mathbf{M}_{-(\mathbf{x},\mathbf{y})} - \mathbf{M}_{-(\hat{\mathbf{x}},\hat{\mathbf{y}})} \right) \mathbf{g}(\hat{\mathbf{x}},\hat{\mathbf{y}}) \right\|_2^2 \quad (18)$$
$$+ \frac{3\eta^2}{\sigma^2} \frac{1}{(n-1)^2} \hat{\mathbb{E}}_{(\mathbf{x},\mathbf{y}) \in \mathcal{D}} \left\| \mathbf{M}_{-(\mathbf{x},\mathbf{y})} \left( \mathbf{g}(\mathbf{x},\mathbf{y}) - \hat{\mathbb{E}}_{(\hat{\mathbf{x}},\hat{\mathbf{y}}) \in \mathcal{D}} \mathbf{g}(\hat{\mathbf{x}},\hat{\mathbf{y}}) \right) \right\|_2^2$$

$$\leq \frac{3\eta^2}{\sigma^2} \hat{\mathbb{E}}_{(\mathbf{x},\mathbf{y}) \in \mathcal{D}} \left( \frac{n}{n-1} \right)^2 \hat{\mathbb{E}}_{(\mathbf{x},\mathbf{y}) \in \mathcal{D}} \left\| \left( \mathbf{M}_{-(\mathbf{x},\mathbf{y})} - \hat{\mathbb{E}}_{(\hat{\mathbf{x}},\hat{\mathbf{y}}) \in \mathcal{D}} \mathbf{M}_{-(\hat{\mathbf{x}},\hat{\mathbf{y}})} \right) \mu(\mathcal{D}) \right\|_2^2$$
$$+ \frac{3\eta^2}{\sigma^2} \frac{1}{(n-1)^2} \hat{\mathbb{E}}_{(\mathbf{x},\mathbf{y}) \in \mathcal{D}} \left\| \mathbf{M}_{-(\mathbf{x},\mathbf{y})} \left( \mathbf{g}(\mathbf{x},\mathbf{y}) - \hat{\mathbb{E}}_{(\hat{\mathbf{x}},\hat{\mathbf{y}}) \in \mathcal{D}} \mathbf{g}(\hat{\mathbf{x}},\hat{\mathbf{y}}) \right) \right\|_2^2 + \mathcal{O} \left( \beta/n^4 \right) \quad (19)$$

Equation (19) uses assumptions of bounded gradient norms and stability of the pre-conditioner matrix to bound the term in Equation (18). $\square$

### D.3 CAN WE CONNECT GRACE'S PREDICTIVE BEHAVIOR OF STUDENT PERFORMANCE TO STUDENT'S VALIDATION LOSS AFTER TRAINING?

In Lemma 1, we establish a connection between leave-one-out conditional mutual information (LOO-CMI) and GRACE. Because LOO-CMI has been used to derive generalization bounds for training algorithms, primarily measured via the model's loss after training, we ask whether GRACE correlates with the student's loss after being trained on different teachers. To do so, we investigate two key questions: (a) does validation loss of the student after training correlate with student performance? and (b) does GRACE correlate with the validation loss of the student after training?

### D.3.1 DOES VALIDATION LOSS AFTER TRAINING CORRELATE TO STUDENT PERFORMANCE?

For each teacher, we create validation set using the teacher's responses on the validation set of prompts. We use the following loss functions of the student after training on a teacher's responses on a training set:

- **Validation loss** is computed as the cross entropy loss of the student on the teacher's responses on the validation set.
- **Validation - Train loss** is computed as the difference in the cross entropy loss of the student on the teacher's responses on the training set and the validation set.
- **Forward KL** is computed as the difference in the cross entropy loss of the student and the teacher on the teacher's responses on the training set.
- **Backward KL** is computed as the difference in the cross entropy loss of the teacher and the student on the student's responses on the training set.

- **Loss on pre-training** is computed as the cross entropy loss of the student on the pre-training data. Because we don't have access to pre-training data of LLaMA-1B, we use examples from fineweb-edu as a surrogate (Lozhkov et al., 2024).

We measure correlations between each loss score and the student average-at-16 performance after training across all teachers.

| Performance score (after training) | Absolute Spearman ($\uparrow$) | Regret ($\downarrow$) |
|---|---|---|
| Validation Loss | 0.15 | 4.1 % |
| Validation Forward KL | 0.44 | 4.6 % |
| Validation Backward KL | 0.14 | 12.9 % |
| Validation - Train Loss | 0.17 | 4.1 % |
| Loss on pre-training | 0.07 | 6.5 % |

Table 1: Setting: LLaMA-1B training on GSM8K. **Observation: None of the loss variants of the student after training correlate with the student performance.** The performance scores have been computed on the student after training on each teacher generations.

In Table 1, we observe that none of the loss-based scores correlate well with student performance. Several factors may explain this. First, the validation sets differ across teachers due to variations in their chain-of-thought reasoning styles, making the resulting losses across teachers non-comparable. More critically, there exists an inherent mismatch between the cross-entropy loss and the auto-regressive generation objective: cross-entropy loss measures the average token-level prediction error on a fixed reference set, whereas auto-regressive evaluation tests whether the model can generate the correct solution through sequential token generation. Thus, a student can have low loss on the teacher's generations after training, but can struggle with getting correct auto-regressive generations. This discrepancy has been extensively discussed in prior works (Arora et al., 2022; Fang et al., 2025).

### D.3.2 DOES GRACE CORRELATE WITH VALIDATION LOSS AFTER TRAINING?

Here, we study whether GRACE correlates with validation loss of the student after training. To do so, we consider the correlations between the loss-based scores of the student after training, described in the previous section, and the scores GRACE, G-Vendi, and G-Norm computed on the gradients of the student before training.

| Performance score (after training) | G-Norm | G-Vendi | GRACE |
|---|---|---|---|
| Validation Loss | 0.34 | 0.27 | 0.09 |
| Validation Forward KL | 0.10 | 0.28 | 0.15 |
| Validation Backward KL | 0.09 | 0.40 | 0.01 |
| Validation - Train Loss | 0.0 | 0.08 | 0.23 |
| Loss on pre-training data | 0.03 | -0.07 | 0.11 |

Table 2: Setting: LLaMA-1B training on GSM8K. Here, we measure absolute spearman correlations between each loss-based scores with each of G-Vendi, GRACE, and G-Norm. **Observation: GRACE doesn't correlate well with any of the loss variants of the student after training.** GRACE, G-Vendi, G-Norm have been computed using the gradients of the student before training, and the performance scores have been computed on the student after training.

As shown in Table 2, GRACE exhibits only a weak correlation with the student's loss across teachers. Interestingly, G-Vendi and G-Norm show even stronger correlation with the student's loss-based scores. This suggests that the predictive effectiveness of GRACE cannot be directly attributed to the student's validation loss after training.

Together, these gaps weaken the theoretical link between LOO-CMI and GRACE, leaving open the question of why GRACE correlates more strongly with student performance. Furthermore, since LOO-CMI only offers an upper bound on the generalization error, it cannot serve as a direct comparative metric. A deeper theoretical understanding of how GRACE relates to student performance after training remains an important avenue for future investigation.

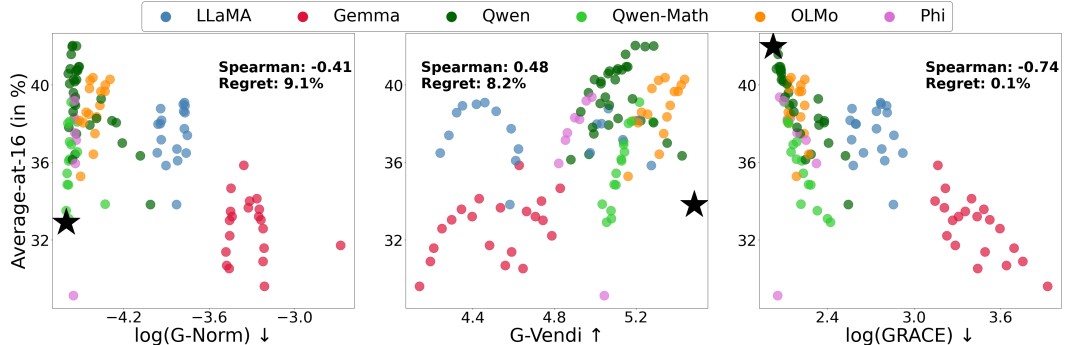

Figure 10: **GRACE achieves** $74\%$ **Spearman correlation to OLMo-1B's post-distillation performance on GSM8K**, significantly outperforming G-Norm (41%) and G-Vendi (48%). When evaluated by teacher-student regret, GRACE selects a teacher with regret of $0.1\%$, outperforming G-Norm and G-Vendi, which incur regrets of $9.1\%$ and $8.2\%$, respectively. Stars denote students trained from the teacher chosen by each score. Similar observations hold for a Gemma-2B student (Figure 11).

## E   MORE EXPERIMENTAL RESULTS AND COMPARISONS

| | Student | Select by GRACE | | Select by Teacher Performance | |
|---|---|---|---|---|---|
| | | Teacher chosen | Student's Performance | Teacher chosen | Student's Performance |
| GSM8K | LLaMA-1B | Qwen2.5-3B-Instruct ($\tau = 0.9$) | **52.2** | LLaMA3.3-70B-Instruct ($\tau = 0.8$) | 44.8 |
| | OLMo-1B | Qwen2.5-3B-Instruct ($\tau = 0.9$) | **46.0** | LLaMA3.3-70B-Instruct ($\tau = 0.8$) | 36.1 |
| | Gemma-2B | Qwen2.5-3B-Instruct ($\tau = 0.6$) | **69.0** | LLaMA3.3-70B-Instruct ($\tau = 0.8$) | 62.4 |
| MATH | LLaMA-3B | Phi-4 ($\tau = 0.8$) | **36.0** | Qwen2.5-7B-Instruct ($\tau = 0.4$) | 34.0 |

Table 3: Summary of student performance (Average-at-16) under teachers selected with GRACE and teacher performance from Figures 2, 10, 11 and 14.

### E.1   ADDITIONAL RESULTS

**GSM8K:**  First, we show the behavior of Gemma-2B, when trained with different teachers in Figure 11. We also observe that GRACE has higher Spearman correlation to the student performance, compared to G-Norm and G-Vendi.

We also report the performance when we allow more computation for the computation of GRACE. In Figures 12 and 13, we show that the final Spearman correlation improves by atleast 7% and 4% for OLMo-1B and LLaMA-1B respectively on GSM8K, when we increase the projected dimension of gradients from 512 to 1024.

**MATH:**  In Figure 14, we show the scatter plot of the performance of the model when trained with different teachers. We show that G-Norm and GRACE achieve 89% and 88% Spearman correlation to the student performance post distillation training, while G-Vendi only achieves 74% Spearman correlation.

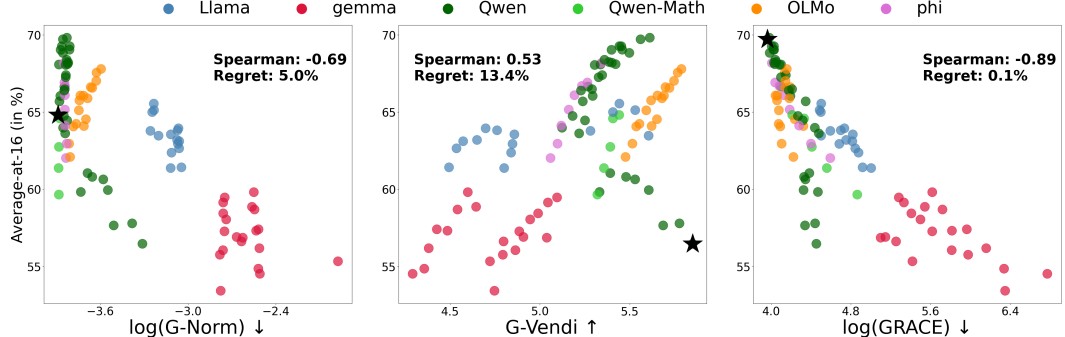

Figure 11: **GRACE achieves $89\%$ correlation to Gemma-2B performance after training on GSM8K, across all teacher, generation temperature combinations.** G-Norm and G-Vendi can achieve 69% and 53% correlation respectively. Here, $n = 512, d = 512$ are used to compute all scores. When evaluated by regret, GRACE selects a teacher that trains the student to within $0.1\%$ of the best achievable performance, outperforming G-Norm and G-Vendi, which incur regrets of $5\%$ and $13.4\%$, respectively.

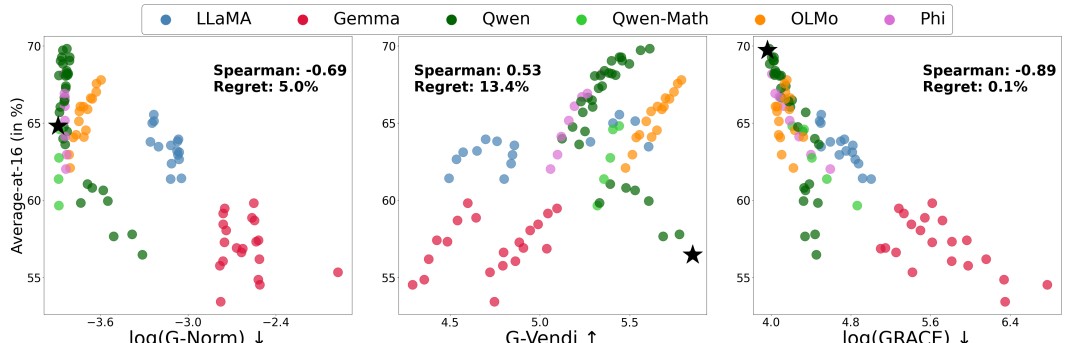

Figure 12: Repeated experiment from Figure 2 but with $d = 1024$. **GRACE achieves $90\%$ correlation to LLaMA-1B performance after training on GSM8K, across all teacher, generation temperature combinations.** G-Norm and G-Vendi can only achieve 53% and 47% correlation respectively. When evaluated by regret, GRACE selects a teacher that trains the student to within $0.3\%$ of the best achievable performance, outperforming G-Norm and G-Vendi, which incur regrets of $10.8\%$ and $14.5\%$, respectively.

## E.2  Measuring effectiveness of GRACE within each teacher family

We measure the regret of teacher selection using each score within individual teacher families. For a given family, the regret of a score is defined as the performance gap between (i) the best student performance achieved after distilling from all teachers in that family and (ii) the student performance obtained when the teacher is chosen according to that score. On average, GRACE yields only $1.04\%$ regret, while other metrics incur at least $3\%$. See Figure 15 for details.

## E.3  Out-of-distribution evaluation

One question is whether GRACE can also predict the performance of the trained teacher in out-of-distribution settings. To check, we measure the performance of the trained LLaMA-1B models (trained on GSM8K) on GSM-Symbolic (Mirzadeh et al., 2025) and measure the Spearman correlations and regrets of G-Norm, G-Vendi, and GRACE with respect to the OOD performance (Figure 16). Similarly, we perform OOD evaluation of LLaMA-3B models (trained on MATH) on MATH$^2$ (Shah et al., 2024) and MATH Perturb (Simple & Hard) (Huang et al., 2025b), and measure the Spearman correlations and regrets of G-Norm, G-Vendi, and GRACE with respect to the OOD performance (Table 4). We observe the following:

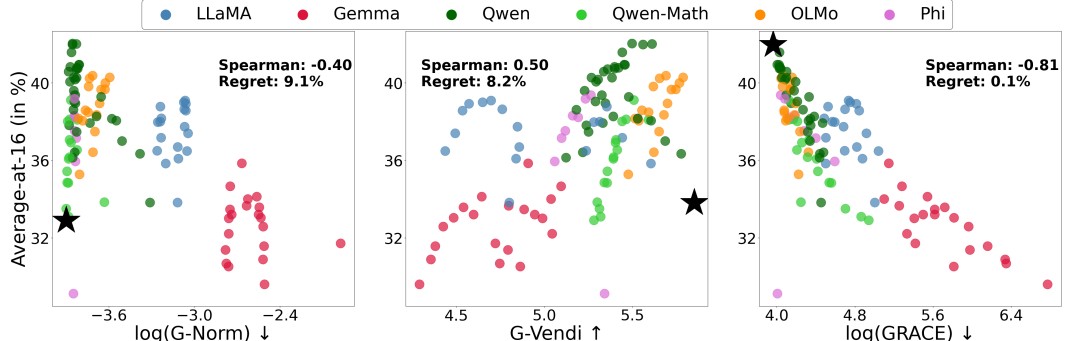

Figure 13: Repeated experiment from Figure 10 but with $d = 1024$. **GRACE achieves** $81\%$ **correlation to OLMo-1B performance after training on GSM8K, across all teacher, generation temperature combinations.** G-Norm and G-Vendi can only achieve 40% and 50% correlation respectively. When evaluated by regret, GRACE selects a teacher that trains the student to within $0.1\%$ of the best achievable performance, outperforming G-Norm and G-Vendi, which incur regrets of $9.1\%$ and $8.2\%$, respectively.

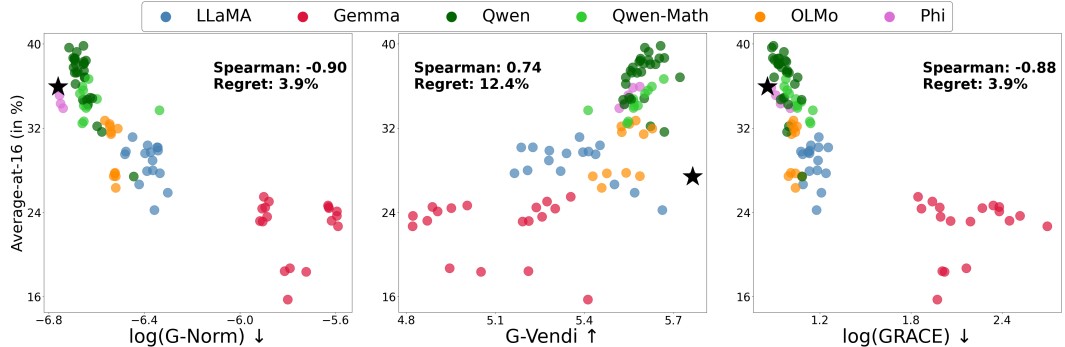

Figure 14: **GRACE achieves** $88\%$ **correlation to LLaMA-3B performance after training on MATH, across all teacher, generation temperature combinations.** G-Norm and G-Vendi can achieve 90% and 74% correlation respectively. Here, $n = 512, d = 512$ are used to compute all metrics. When evaluated by regret, GRACE selects a teacher that trains the student to within $3.9\%$ of the best achievable performance (similar to G-Norm), outperforming G-Vendi, which incurs regret of $12.4\%$.

- On LLaMA-1B, GRACE has both high Spearman correlation ($78\%$) and low regret ($0.8\%$) compared to G-Norm and G-Vendi. Thus, good in-distribution performance of the student also translates to good OOD performance.

- On LLaMA-3B, we observe a more nuanced pattern. G-Norm attains higher Spearman correlation and lower regret than GRACE on MATH Perturb (Simple), which constitutes a relatively mild out-of-distribution shift. In contrast, GRACE matches or surpasses G-Norm on the more challenging MATH$^2$ and MATH Perturb (Hard) settings. These results suggest that GRACE provides a reliable indicator of student performance under substantial distribution shifts.

### E.4 HOW DOES KNOWING THE TEACHER PERFORMANCE CHANGE THE BEHAVIOR OF THE DIFFERENT SCORES?

Our main results focus on the setting where no verifier is used to pre-filter the teachers. A key question is whether the teacher's own performance influences the correlation between GRACE and student performance, and if teacher performance were known, whether it could further improve the predictive power of various scores. To examine this, we analyze how the behavior of GRACE varies when teachers are grouped into different performance bins.

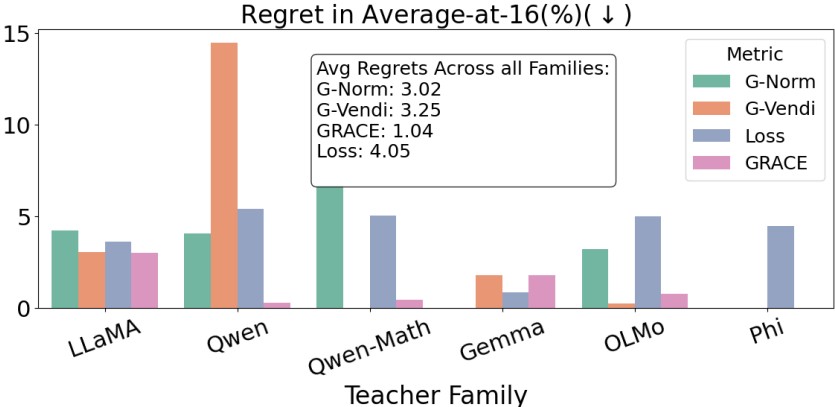

Figure 15: Here, we measure the regret incurred when a score is used to select a teacher within each teacher family. On average, GRACE identifies a teacher that enables the student to achieve performance within $1.04\%$ of the best performance possible from distillation with a teacher in that family, whereas other scores incur an average regret of at least $3\%$.

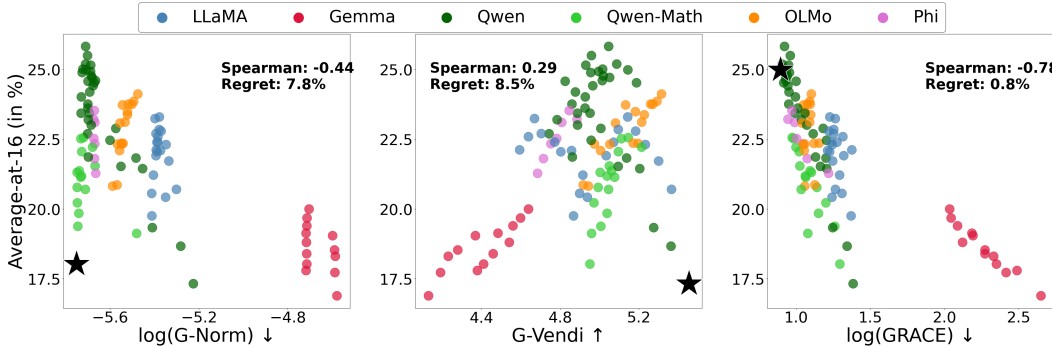

Figure 16: **Out-of-distribution results show consistent trends on GSM8K.** We further evaluate the LLaMA-1B students (trained on GSM8K) on the GSM-Symbolic dataset (Mirzadeh et al., 2025), an out-of-distribution benchmark relative to GSM8K. GRACE achieves a Spearman correlation of $78\%$ with student performance, outperforming both G-Norm and G-Vendi. As expected, this is slightly lower than the $86\%$ correlation observed in-distribution (Figure 2). In terms of regret, GRACE achieves the lowest regret ($0.8\%$), outperforming both G-Norm and G-Vendi in this OOD setting.

To verify, we divide all our teachers into three equal bins based on their own performance. In each bin, we compute the Spearman correlation of each metric to the student performance post distillation training. We also report the regret of a score within each bin as the performance gap between (i) the best student performance achieved after distilling from all teachers in that performance bin and (ii) the student performance obtained when the teacher is chosen according to that score.

- In Figure 17, we show that GRACE shows better Spearman correlation and also lower regret, across all the performance bins for LLaMA-1B training on GSM8K. Advantage of GRACE are particularly more pronounced for distinguishing high performing teachers.
- In Figure 18, for LLaMA-3B training on MATH, GRACE maintains stronger Spearman correlation with student performance across all bins. However, G-Vendi shows lower regret than GRACE in the highest-performance teacher bin.

## E.5 MEASURING CORRELATIONS WITH OTHER MEASURES OF STUDENT PERFORMANCE

We primarily measured the correlation of the metrics to Average-at-16 performance of the student post distillation, when we generate responses at temperature $1.0$. A key question is: how do the correlations change when the performance metric of the student is changed? To verify, we use two other commonly used performance metrics – greedy student performance and Best-of-16 performance.

| Evaluation dataset | Best Student Average-at-16 | Score | Absolute Spearman ($\uparrow$) | Regret ($\downarrow$) |
|---|---|---|---|---|
| MATH Perturb (Simple) | 31.32% | G-Norm | **0.89** | **2.8** % |
| | | G-Vendi | 0.59 | 16.7% |
| | | GRACE | 0.76 | **2.8** % |
| MATH$^2$ | 9.36% | G-Norm | 0.87 | **0.1** % |
| | | G-Vendi | 0.66 | 3.9% |
| | | GRACE | **0.88** | **0.1**% |
| MATH Perturb (Hard) | 13.72% | G-Norm | 0.80 | **2.1** % |
| | | G-Vendi | 0.75 | 5% |
| | | GRACE | **0.88** | **2.1**% |

Table 4: **Out-of-distribution results show consistent trends on MATH.** We further evaluate the LLaMA-3B trained students (on MATH) on MATH$^2$ (Shah et al., 2024) and MATH Perturb (Simple & Hard) (Huang et al., 2025b), out-of-distribution benchmarks relative to MATH. While G-Norm yields similar or higher Spearman correlation and lower regret than GRACE on MATH Perturb (Simple), a mild OOD benchmark, GRACE outperforms G-Norm on both MATH$^2$ and MATH Perturb (Hard), which represent substantially more challenging OOD shifts.

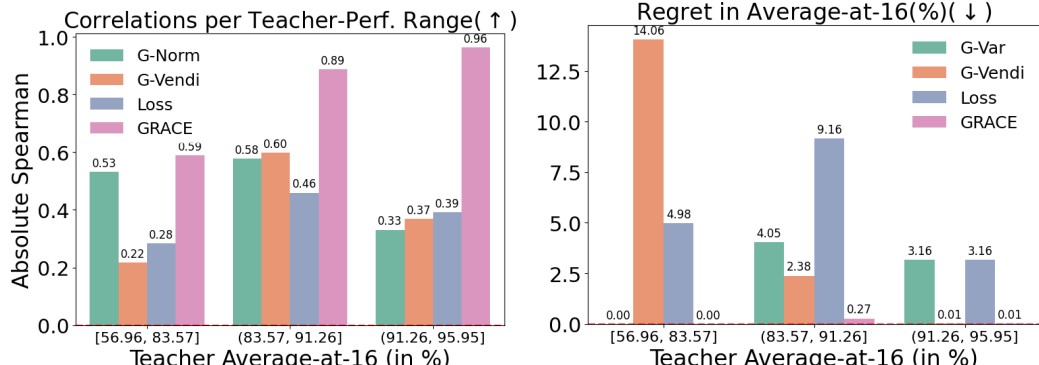

Figure 17: Setting: LLaMA-1B training on GSM8K. When we **divide teachers into three equal bins** based on their own performance, we compare both **Spearman correlation** and **Regret**. Across all bins, GRACE shows a stronger Spearman correlation with student performance after distillation and consistently shows minimum regret. Its advantage is more pronounced when distinguishing among high-performing (top performance bin) teachers.

Greedy performance measures correctness of single student's generation response per prompt at temperature 0.0. Best-of-16 performance measures whether the student gets atleast one correct response, when generating 16 responses per prompt at temperature 1.0. In Table 5, we observe that the Spearman correlation of GRACE to the student performance drops, when we switch performance measure to greedy accuracy or Best-of-16 performance. However, GRACE still achieves small regret with the new performance metrics, compared to G-Norm and G-Vendi.

| Metric | Average-at-16 ($\tau = 1.0$) | | Greedy accuracy | | Best-of-16 ($\tau = 1.0$) | |
|---|---|---|---|---|---|---|
| | Abs. Spearman ($\uparrow$) | Regret ($\downarrow$) | Abs. Spearman ($\uparrow$) | Regret ($\downarrow$) | Abs. Spearman ($\uparrow$) | Regret ($\downarrow$) |
| G-Vendi | 0.44 | 14.5 % | 0.65 | 4.5% | **0.68** | 4.5% |
| G-Norm | 0.55 | 4.9 % | 0.33 | 13.5% | 0.30 | 7.5% |
| GRACE | **0.86** | **0.3**% | **0.70** | **2.4**% | 0.64 | **0**% |

Table 5: **GRACE leads to stronger students in terms of various performance measures.** We compare G-Vendi, G-Norm, and GRACE using average-at-16, greedy, and best-of-16 performance of the post-distillation students. Results are on LLaMA-1B training on GSM8K (setting of Figure 2). GRACE achieves the strongest or close to stronger Spearman correlation, and consistently leads to the lowest regret compared to G-Norm and G-Vendi.

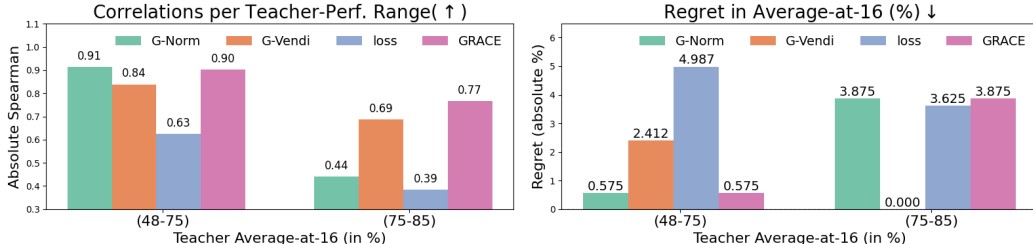

Figure 18: Setting: LLaMA-3B training on MATH. We split teachers into two equal bins according to their own performance and evaluate each bin using both **Spearman correlation** and **regret**. Across bins, GRACE maintains a stronger correlation with the student's post-distillation performance. However, in the top-performing bin, G-Vendi achieves lower regret than GRACE.

| Student Temperature | Absolute Spearman (↑) | | | Regret (↓) | | |
|---|---|---|---|---|---|---|
| | G-Norm | G-Vendi | GRACE | G-Norm | G-Vendi | GRACE |
| 0.0 | 0.34 | 0.63 | **0.70** | 13.5 | 4.5 | **2.4** |
| 0.3 | 0.42 | 0.65 | **0.75** | 9.2 | 4.1 | **0.2** |
| 0.6 | 0.51 | 0.62 | **0.83** | 9.5 | 8.5 | **1.2** |
| 0.8 | 0.58 | 0.57 | **0.87** | 8.0 | 8.3 | **0.3** |
| 1.0 | 0.44 | 0.56 | **0.86** | 10.8 | 14.5 | **0.3** |

Table 6: **GRACE can guide temperature choice.** We compare G-Norm, G-Vendi, GRACE in terms of the student's average-at-16 accuracy at various temperatures. Results are on LLaMA-1B training on GSM8K (setting of Figure 2). GRACE achieves the highest Spearman correlation (0.7 and above) and the lowest regret across all temperatures, demonstrating robustness to the temperature choice.

### E.6 PRELIMINARY EXPLORATION: GENERAL NLP TASKS BEYOND MATH

One question is whether the predictive nature of GRACE holds beyond math based tasks. We study ARC (Clark et al., 2018). ARC is a general reasoning task, where the model is given a question and 4 choices to choose from to answer. An example of a question in ARC is as follows:

> A fold observed in layers of sedimentary rock most likely resulted from the:
> (a) *cooling of flowing magma.*
> (b) *converging of crustal plates.*
> (c) *deposition of river sediments.*
> (d) *solution of carbonate minerals.*

**Experiment Setting:** We train a LLaMA-1B model and use the same set of teachers as in our GSM8K and MATH experiments, considering a subset of generation temperatures $0.4, 0.6, 0.8, 1.0$. Each teacher is prompted to provide the correct answer explicitly, rather than just the choice number. Nonetheless, some teachers still respond with the choice number only. To account for this, our evaluation accepts either the choice number or the full answer. For choice numbers, we apply exact match, while for textual answers, we compute the edit distance to each of the four options to determine correctness. We sample 16 responses per prompt from each teacher to construct the training dataset and evaluate performance on the ARC validation split.

**Primary observations:** We observe that the teacher's performance itself is a strong indicator of the student performance after training. However, among all scores that do not assume access to such verifier based filtering on the teacher, GRACE returns the best correlation to the student performance (however, lower than Spearman correlation it could achieve on math based tasks). Among all scores (teacher performance included), GRACE returns lower regret in selecting the best teacher for training the student.

These observations indicate that GRACE is not a perfectly predictive measure of student performance after training. However, it remains highly effective for selecting strong teachers, consistently yielding lower regret compared to other metrics. For instance, in our experiments, Gemma-2B teachers exhibited much higher (worse) GRACE scores ($10^{-3}$) than Qwen-Math models ($10^{-4}$), yet students distilled from Gemma-2B teachers ($56.4\%$) outperformed those trained from Qwen-Math ($41.0\%$).

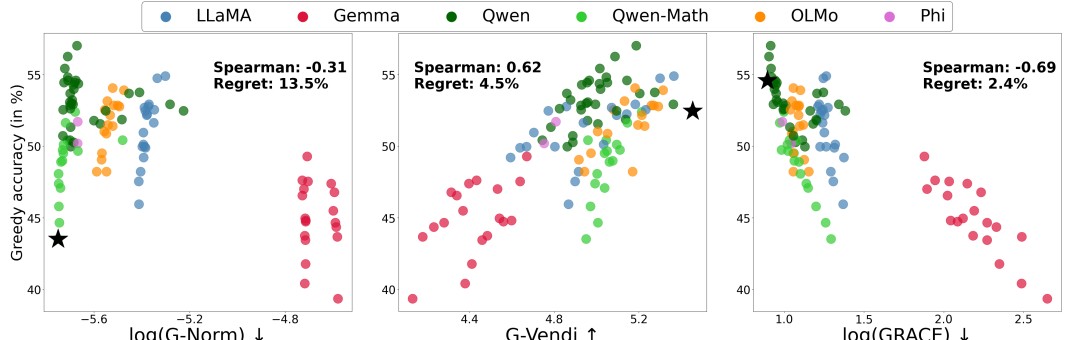

Figure 19: Repeated experiment from Figure 2 but greedy performance of trained student model. **GRACE achieves only** $70\%$ **correlation to LLaMA-1B performance after training on GSM8K, across all teacher, generation temperature combinations.** This is a sharp reduction from $86\%$ correlation to Average-at-16. However, GRACE still predicts close to the optimal teacher.

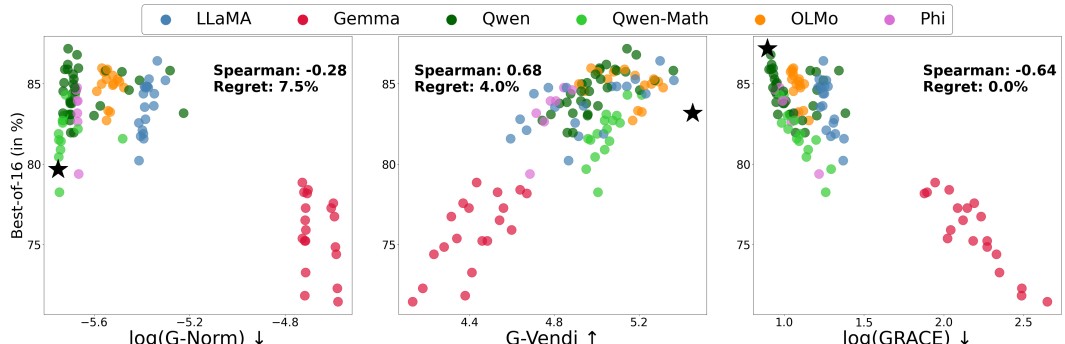

Figure 20: Repeated experiment from Figure 2 but best-of-16 performance of trained student model. **GRACE achieves only** $65\%$ **correlation to LLaMA-1B performance after training on GSM8K, across all teacher, generation temperature combinations.** This is a sharp reduction from $86\%$ correlation to Average-at-16. However, GRACE still predicts the optimal teacher.

Conversely, directly choosing the highest-performing teacher (Llama-70B-instruct) led to a student with an average-at-16 score of $58.6\%$, whereas selecting based on GRACE produced a student reaching $59.5\%$.

This pattern stands in contrast to our findings on math-related tasks, where GRACE was highly predictive of student performance after distillation and also could reliably identify close-to-optimal teacher. We keep extensive search beyond math-related tasks to future work.

> Overall, **GRACE remains a reliable score for teacher selection**, even though teachers with low GRACE values may yield strong students—leading to slightly weaker correlations. Identifying the conditions under which GRACE serves as a near-perfect predictor of student performance remains an important open question for future work.

### E.7 COMPUTATIONAL COMPLEXITY

GRACE is computationally inexpensive to compute. As shown in Table 8, for $n = d = 512$ and $m = 4$, the gradients for each model takes around 10 minutes to compute and around 4.3MB to store. Computing the GRACE score given the gradients of the student w.r.t. a teacher is inexpensive and takes roughly 10 seconds.

| Score | Absolute Spearman ($\uparrow$) | Regret ($\downarrow$) (Best Average-at-16 student: 61%) |
|---|---|---|
| G-Vendi | 0.11 | 19.1 % |
| G-Norm | 0.43 | 2.4 % |
| GRACE | 0.56 | **1.5**% |
| Determinant | 0.09 | 28.9 % |
| BADGE | 0.19 | 28.9 % |
| Gradient inner product | 0.09 | 6.4 % |
| Gradient cosine | 0.19 | 28.9 % |
| Student's loss on teacher's generations | 0.14 | 12.9% |
| Average (token) probabilities | 0.09 | 11.5% |
| Average length | 0.06 | 6.4% |
| Teacher average-at-16 performance | **0.82** | 2.4 % |

Table 7: LLaMA-1B training on ARC. **Among all scores that do not rely on a verifier for teacher filtering, GRACE achieves the best Spearman correlation and regret in identifying the optimal teacher.** While teacher performance, that assumes access to a verifier, shows much stronger correlation with student performance after training, GRACE still yields lower regret and remains the more effective criterion for teacher selection.

| | Gradient Features Computation | Metric Computation |
|---|---|---|
| Computation complexity Running time | $\mathcal{O}(n \cdot m \cdot P \cdot d)$ $\approx 10$ minutes | $\mathcal{O}(n \cdot m \cdot d^2 + d^3)$ $< 10$ seconds |
| Storage Complexity Actual storage | $\mathcal{O}(n \cdot m \cdot d)$ 4.3 MB | - - |

Table 8: Time complexity to compute GRACE. The running time and the actual storage have been computed on $\tilde{n} = 512$, $m = 4$, $d = 512$ for LLaMA-1B training on GSM8K, and have been reported as a rough average across all settings. Wall-clock time has been reported on a single H100 (80 GB) GPU. For gradient computation, we use 32 parallel CPU threads following Park et al. (2023). Here, $P$ denotes the number of parameters in the model.

## F ABLATIONS

### F.1 FILTERING V/S NO FILTERING

In our experiments in the main paper, we perform no filtering of the responses from the teacher. Here, we compare to the case when we filter the teacher's responses by correctness. We sample 16 responses from each teacher and remove the incorrect responses. Then, we sample with repetition to get a set of 16 responses to train the model.

First, we find that the student gets worse performance with filtering of correct responses from the teacher (Figure 22). However, we find that when we compare our metrics to the student performance after training, we find that our metrics have slightly higher spearman correlation with the student performance when we train with filtering on teacher responses, compared to student trained with no filtering on the teacher responses (Figure 23).

### F.2 ABLATION ON TRAINING HYPERPARAMETERS

We observe that a Llama-1B model trained on generations of Llama-70B Instruct models and Gemma-2-27B Instruct models perform badly. We train with learning $1e0^{-5}$ on the 16 generations per prompt of the teacher for 4 epochs. One primary question is whether the small model is over-optimizing on the teacher's generations. To check this, we track the train and test performance of the trained model with varying number of generations (Figure 24) and epochs of training (Figure 25). We observe that the performance of the trained student model improves with increasing number of epochs and number of generations, implying no over-optimization in our training setting.

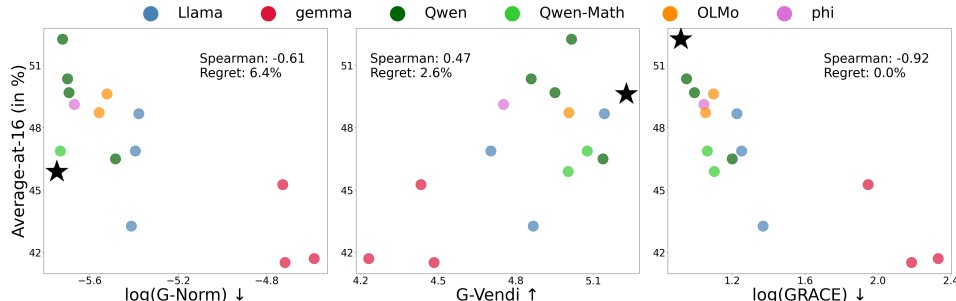

Figure 21: **Qualitative study: GRACE can effectively correlate with student performance when compared across different teacher choices.** Here, we report LLaMA-1B performance on GSM8K across different teacher choices at a generation temperature 0.6. G-Norm can capture differences between Llama and Qwen teacher families, however can't differentiate between Qwen, Qwen-Math and Phi teacher families. G-Vendi provides better separation among these families, but assigns higher scores to sub-optimal teachers. GRACE achieves 90% correlation with student performance after training, while also predicting Qwen-3B-Instruct to be the optimal teacher.

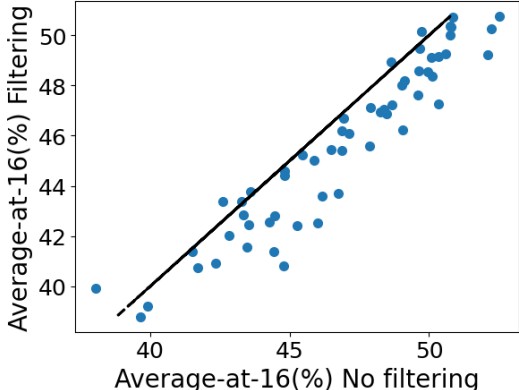

Figure 22: Comparing teachers, when we filter correct responses from the teacher v/s when we don't filter correct responses from the teacher. Here, we train LLaMA-1B on GSM8K with 15 teachers and generation temperatures 0.4, 0.6, 0.8, 1.0. We compare students trained from teacher without filtering (x-axis) with students trained from teacher with correct answer filtering (y-axis). We find that students trained with no filtering outperforms models trained with filtering.

### F.3 ABLATIONS ON THE PARAMETERS OF GRACE

In Figure 26, we use LLaMA-1B trained on GSM8K as a case study and ablate two key factors affecting GRACE: the gradient projection dimension ($d$) and the number of prompts ($n$) used to compute the score. With a fixed $n$, the correlation between GRACE and student performance consistently improves as $d$ increases. Conversely, when holding $d$ fixed, correlation increases with $n$, peaking when $n = d/2$.

We further vary the number of generations per prompt ($m$) while keeping $n$ and $d$ fixed. In our base configuration, using $m = 2, 4, 8$ yields correlations of $0.70$, $0.86$, and $0.87$, respectively, suggesting that $m = 4$ already provides a sufficiently strong estimate of GRACE.

We additionally vary the number of cross-validation splits used in GRACE. As shown in Figure 27, the correlations to the student performance do not vary much for both GSM8k and MATH for more than $C = 6$ splits. We take $C = 10$ as the default.

### F.4 GRADIENT NORM'S RELATION TO LENGTH

Figure 28 shows that the norm of the gradient (averaged over tokens) on a generation decreases as the generation length grows, roughly following a trend of $1/\log T$ for length-$T$ generations, consistent

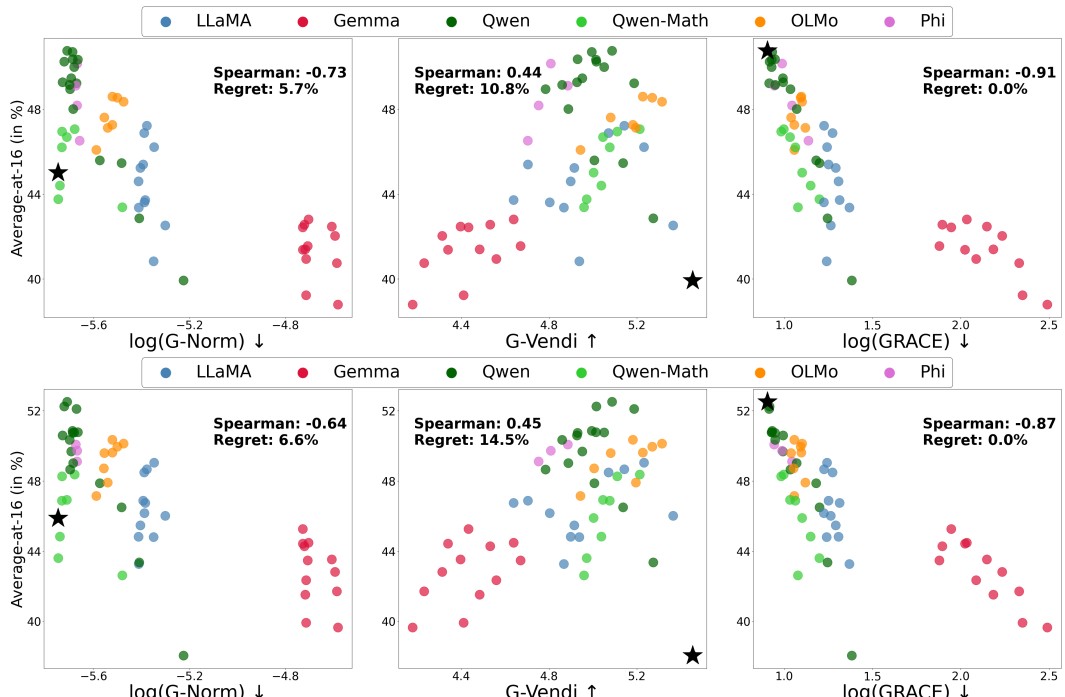

Figure 23: Comparisons between the metrics and the student performance when we filter responses v/s we don't filter correct responses from the teacher. Here, we train LLaMA-1B on GSM8K with 15 teachers and generation temperatures 0.4, 0.6, 0.8, 1.0. We find that our metrics have slightly higher spearman correlation to the student performance when we filter correct responses from the teacher and train only on them.

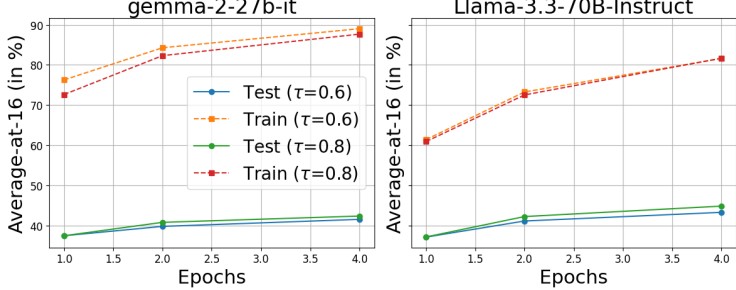

Figure 24: LLaMA-1B training on GSM8K with 16 responses per prompt of gemma-27b-instruct and llama-70b instruct model. We vary the number of epochs and observe that both train and test performance improves with more epochs of training.

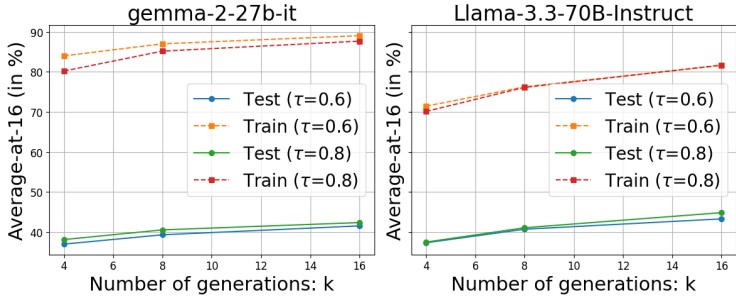

Figure 25: LLaMA-1B training on GSM8K with varying number of responses per prompt of gemma-27b-instruct and llama-70b instruct model. We observe that both train and test performance improves with more training samples from the teacher.

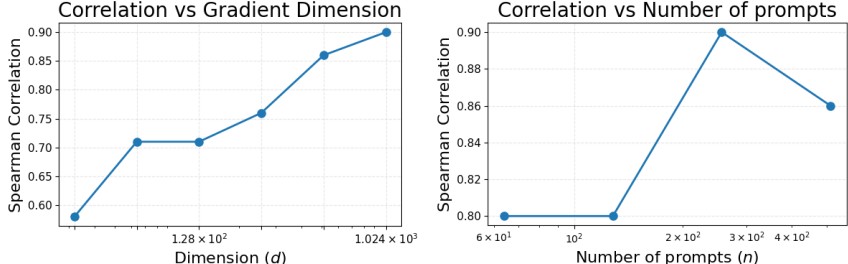

Figure 26: Varying hyperparameters for GRACE on LLaMA-1B training on GSM8K. We use the base setup as $n = 512$, and $d = 512$. We vary one of them, while fixing the other. Main takeaway: (a) GRACE improves with increasing gradient dimension, (b) GRACE generally increases with number of prompts that we consider but shows a small dip as we increase further. $n = d/2$ gives the best results for spearman correlation.

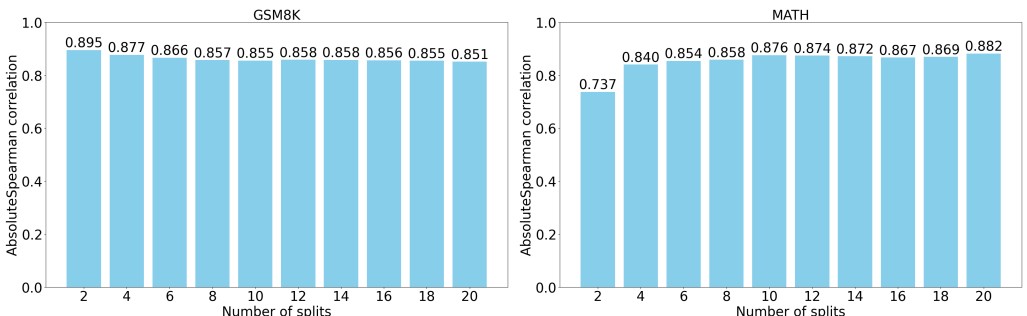

Figure 27: **Varying number of cross-validation splits** on GSM8K (left) and MATH (right).

with observations in Xia et al. (2024). Intuitively, this is likely because longer generations tend to contain a larger fraction of less important tokens that do not contribute much to the overall gradient. This observation motivates the $\log T$ scaling in Section 2.

## F.5 ABLATION ON ROBUSTNESS OF METRICS

We check the robustness of each metric by reporting the distributions of the metric values computed over random subsets of teachers. Specifically, we use 100 random draws of subsets consisting of 60% of teachers.

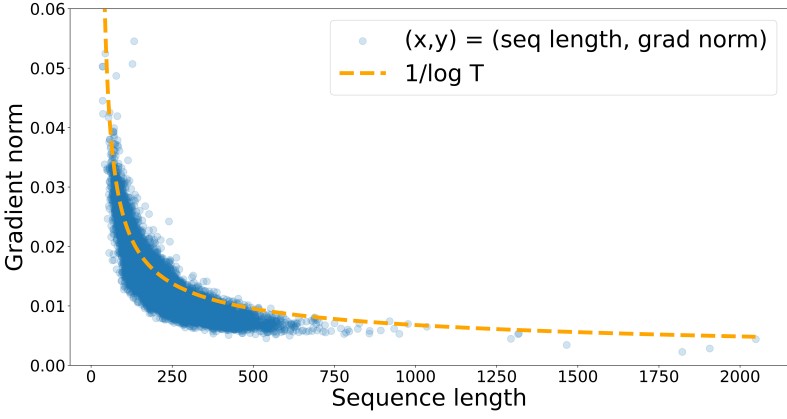

Figure 28: **Gradient norm decreases inversely with** $\log T$, where $T$ is the sequence length. This motivates the gradient scaling in Section 2.

We compare GRACE against the baselines listed in Section C.3, with an additional score of the average probabilities of the correct responses. Among all candidate metrics, GRACE is the only one showing consistently strong correlations.

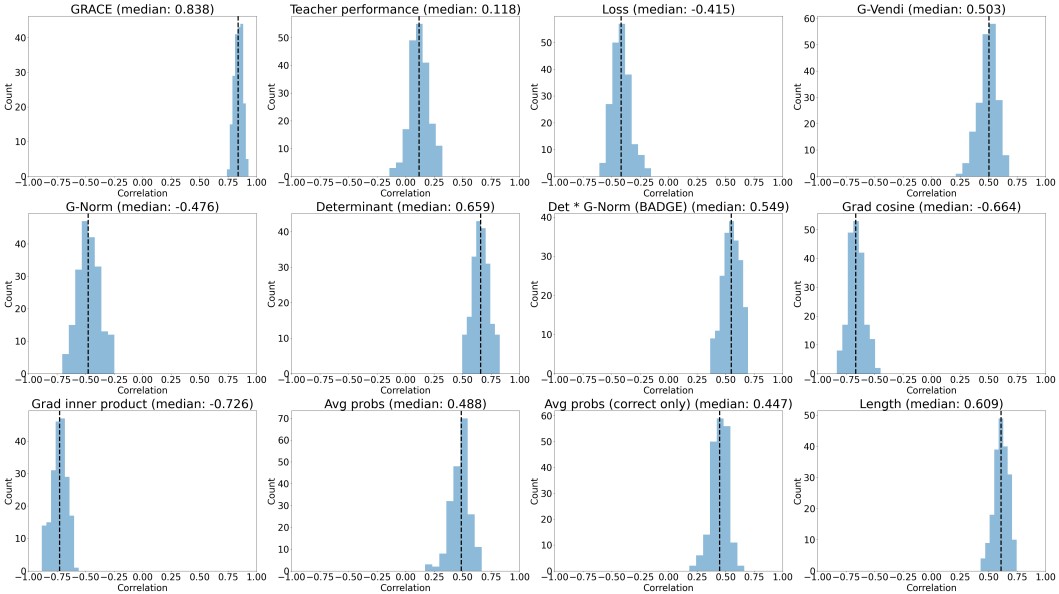

Figure 29: **Robustness of scores**: We report the distribution of scores for LLaMA-1B on GSM8K, computed over 200 random subsets of teacher-temperature combinations, where each subset is 60% of the full set. The dashed lines and numbers in parentheses mark the median. The proposed GRACE score consistently shows stronger correlations over all other metrics.

## G  PROMPT FORMAT

We train the student model with the following format across distillation from all teachers:

### Problem: {input}
### Solution: {output}

One example is as follows:

```
### Problem:  Chandra has four bowls.  Each one is a different
color (red, blue, yellow, green).  She also has exactly one
glass the same color as each bowl.  If she chooses a bowl and
a glass from the cupboard, how many pairings are possible?  One
such pairing is a blue bowl and a yellow glass.
### Solution:

To determine the total number of possible pairings between
   bowls and glasses,
we need to consider the number of choices available for each
   item and then
multiply these choices together.

Chandra has 4 bowls, and each bowl can be paired with any of
   the 4 glasses.
Therefore, the number of possible pairings is calculated by
   multiplying the
number of choices for the bowl by the number of choices for the
   glass.
This can be expressed as:

\\[ 4 \\text{ (choices for bowls)} \\times 4 \\text{ (choices
   for glasses)} = 16 \\]

Thus, the total number of possible pairings is
   \\(\\boxed{16}\\).
% \end{verbatim}
```

We use the following formats to get responses from each teacher family, which have been taken from their model-cards on huggingface. We use vLLM 0.7.0 ((Kwon et al., 2023)) to get responses from all models.

```
Prompt to LLaMA teacher models

<|start_header_id|>system<|end_header_id|>You are a helpful
   assistant.<|eot_id|>
<|start_header_id|>user<|end_header_id|>
Please reason step by step, and put your final answer within
   \\boxed{{}}.
{input}
<|eot_id|>
<|start_header_id|>assistant<|end_header_id|>
```

---

**Prompt to Qwen teacher models**

```
<|im_start|>system\nYou are a helpful assistant.<|im_end|>
<|im_start|>user
Please reason step by step, and put your final answer within
    \\boxed{{}}.
{input}
<|im_end|>
<|im_start|>assistant
```

---

**Prompt to OLMo teacher models**

```
<|endoftext|><|user|>
Please reason step by step, and put your final answer within
    \\boxed{{}}.
{input}
<|assistant|>
```

---

**Prompt to Phi teacher models**

```
<|im_start|>system<|im_sep|>
You are a medieval knight and must provide explanations to
    modern people.
<|im_end|>
<|im_start|>user<|im_sep|>
Please reason step by step, and put your final answer within
    \\boxed{{}}.
{input}
<|im_end|>
<|im_start|>assistant<|im_sep|>
```

---

**Prompt to Gemma teacher models**

```
<bos><start_of_turn>user
You are a helpful assistant.
Please reason step by step, and put your final answer within
    \\boxed{{}}.
{input}
<end_of_turn>
<start_of_turn>model
```

---

### G.1 QUALITATIVE ANALYSIS OF RESPONSES FROM DIFFERENT TEACHERS

Here, we give qualitative analysis of responses from each teacher. In the table below, we show the average length of responses from different teachers. We observe that Gemma models give very concise responses, which leads to very high G-Norm and GRACE (in Figure 2). Response length can differentiate between the responses of different teacher families, however, it's not a perfect predictor (Figures 3 and 4).

We give examples of responses from Qwen-2.5 3B Instruct, LLaMA-3.3 70B Instruct, and Gemma-2 9B Instruct teachers to a question. We find that even though both Qwen-2.5 3B Instruct and LLaMA-3.3 70B Instruct qualitatively give similar amount of details in each step in their responses, Qwen teachers provide more details in their numerical computations.

| Teacher model | Average response length (across all temperatures) |
|---|---|
| **LLaMA** | |
| LLaMA-3.1-8B Instruct | 212.083 |
| LLaMA-3.2-3B Instruct | 224.492 |
| LLaMA-3.3-70B Instruct | 222.314 |
| **Qwen** | |
| Qwen2.5-1.5B Instruct | 259.411 |
| Qwen2.5-3B Instruct | 289.952 |
| Qwen2.5-7B Instruct | 271.666 |
| Qwen2.5-14B Instruct | 287.611 |
| **Qwen-Math** | |
| Qwen2.5-Math-1.5B Instruct | 272.988 |
| Qwen2.5-Math-7B Instruct | 332.119 |
| **Gemma-2** | |
| Gemma-2-2b-it | 166.888 |
| Gemma-2-9b-it | 138.462 |
| Gemma-2-27b-it | 140.294 |
| **OLMo-2** | |
| OLMo-2-7B Instruct | 286.604 |
| OLMo-2-13B Instruct | 252.683 |
| **Phi** | |
| Phi-4 | 293.160 |

Table 9: Average response lengths (in terms of number of tokens under the tokenizer of LLaMA-1B) across all teachers on GSM8K on responses to a subsampled set of 512 prompts from the training set.

---

**Question**

```
Janet's ducks lay 16 eggs per day. She eats three for breakfast
every morning and bakes muffins for her friends every day
with four. She sells the remainder at the farmers' market daily
for $2 per fresh duck egg. How much in dollars does she make
every day at the farmers' market?
```

---

**Response from Gemma-2-9B Instruct (temperature=$0.6$)**

```
Here's how to solve the problem step-by-step:

1. **Eggs used:** Janet uses 3 eggs for breakfast +
4 eggs for muffins = 7 eggs per day.

2. **Eggs remaining:** She has 16 eggs - 7 eggs = 9 eggs
left for the market.

3. **Earnings:** She sells 9 eggs * $2 per egg = $18.

\\boxed{$18}
```

| Teacher model | Average response length (across all temperatures) |
|---|---|
| **LLaMA** | |
| LLaMA-3.1-8B Instruct | 519.514 |
| LLaMA-3.2-3B Instruct | 537.696 |
| LLaMA-3.3-70B Instruct | 529.951 |
| **Qwen** | |
| Qwen2.5-1.5B Instruct | 572.480 |
| Qwen2.5-3B Instruct | 581.836 |
| Qwen2.5-7B Instruct | 561.407 |
| Qwen2.5-14B Instruct | 587.693 |
| **Qwen-Math** | |
| Qwen2.5-Math-1.5B Instruct | 531.055 |
| Qwen2.5-Math-7B Instruct | 784.821 |
| **Gemma** | |
| Gemma-2-2B Instruct | 357.610 |
| Gemma-2-9B Instruct | 341.838 |
| Gemma-2-27B Instruct | 355.279 |
| **OLMo** | |
| OLMo-2-7B Instruct | 632.502 |
| OLMo-2-13B Instruct | 589.932 |
| **Phi** | |
| Phi-4 | 666.602 |

Table 10: Average response lengths (in terms of number of tokens under the tokenizer of LLaMA-3B) across all teachers on MATH on responses to a subsampled set of 512 prompts from the training set.

---

**Response from LLaMA-3.3-70B Instruct (temperature=$0.6$)**

```
To find out how much Janet makes every day at the farmers' market
, we first need to determine how many eggs she sells there.
Here's the step-by-step process:

1. **Total eggs laid per day**: Janet's ducks lay 16 eggs per day.

2. **Eggs eaten for breakfast**: She eats 3 eggs for breakfast
every morning.

3. **Eggs used for baking muffins**: She uses 4 eggs to bake
muffins for her friends every day.

4. **Total eggs used**: The total number of eggs she
uses (eats and bakes) is 3 (for breakfast) + 4 (for muffins) =
7 eggs.

5. **Eggs sold at the farmers' market**: The number of eggs she
sells at the market is the total number of eggs laid minus
the eggs she uses. So, 16 (total eggs) - 7 (eggs used) =
9 eggs sold.

6. **Money made from selling eggs**: She sells each egg for $2.

Therefore, the amount of money she makes
from selling 9 eggs is 9 eggs * $2/egg = $18.

So, Janet makes $18 every day at the farmers' market.
The final answer is: $\\boxed{18}$
```
43

**Response from Qwen-2.5-3B Instruct (temperature=**0.6**)**

```
To determine how much Janet makes every day at the farmers'
market, we need to follow these steps:

1. Calculate the total number of eggs laid by the ducks per day.
2. Subtract the number of eggs Janet eats for breakfast.
3. Subtract the number of eggs used for baking muffins.
4. The remainder will be the number of eggs sold
at the farmers' market.
5. Multiply the number of eggs sold by the price per egg to find
out how much she makes each day.

Let's go through each step in detail:

1. The ducks lay 16 eggs per day.
2. Janet eats 3 eggs for breakfast.
3. Janet uses 4 eggs for baking muffins.

First, we calculate the number of eggs remaining after
Janet's consumption and baking:
\\[
16 \\text{ (total eggs)} - 3 \\text{ (eaten for breakfast)} -
4 \\text{ (used for baking)} = 9 \\text{ (remaining eggs)}
\\]

Next, Janet sells the remaining 9 eggs at the farmers' market
for $2 per egg. So, we calculate her earnings:
\\[
9 \\text{ (remaining eggs)} \\times 2 \\text{ (dollars per egg)}
= 18 \\text{ (dollars)}
\\]

Thus, Janet makes \\(\\boxed{18}\\) dollars every day at
the farmers' market.
```

