# OpenReview forum: "In Good GRACES: Principled Teacher Selection for Knowledge Distillation"
_ICLR.cc/2026/Conference — ICLR 2026 Poster_

### Official Review · Reviewer_vXv9 · 2025-10-31

**Soundness:** 2
**Presentation:** 1
**Contribution:** 4
**Rating:** 4
**Confidence:** 3

**Summary:**

This paper proposes a metric for evaluating teacher LLMs for synthetic data generation, for training student models. They propose a per-sample student-model gradient based metric GRACE for evaluating the teacher models. The proposed metric is well correlated with student model performance, and often currently identifies the best/ideal teacher for training the student. The metric is evaluated across a range of teacher model families, and multiple student models and datasets.

**Strengths:**

1. The proposed metric is well motivated, and reasonably extends G-Vendi, and is easy to compute cheaply.
1. The proposed method achieves good predictive correlation and succesffully predicts the best teacher model across multiple datasets and student models.
1. The proposed method correctly identifies/predicts generation temperature across different teacher model sizes, unlike some prior baselines which monotonically increase with temperature.

With synthetic LLM-generated data and seqKD being used ever more often, this paper explores a useful direction.

**Weaknesses:**

1. Most of the experiments in the paper (except figure 11) use temperature 1. Most llms however are typically used a much lower temperatures. The performance correlation of GRACE drops sharply for greedy sampling, which implies the method is indeed affected by temperature. This significantly weakens the impact of most of the analysis/experiments in this paper.
1. The proposed method uses an ad-hoc gradient scaling of $1/log(L)$, motivated by a decrease in gradient norm with increasing sequence length, and the authors show the proposed scaling fitting their dataset. But Xia et al 2024 (which the authors refer to) shows in Figure 3 that this gradient shape varies somewhat across datasets, and is difficult to model. In my own experiments, modelling token gradients correlated random variables along the sequence gives good fits ($1/sqrt(L)$) at predicting grad norm, and in some other domains, losses from longer sequences are sometimes normalized with $1/L$. It is not immidiately obvious why this particular proposed gradient scaling would be ideal. This same scaling seems to have also been used for the baselines.
1. While the paper presents a very detailed analysis and results, the paper layout is somewhat confusing. The appendix in particular seems not well laid-out.

**Questions:**

1. Regarding Figure 21, does this show the gradient norms for gsm or for math? How does the gradient norm distribution vary across the two datasets?
1. Did the authors try other fits for the gradient norm, for example power law $G = A + B*L^C$ ?
1. How sensitive is the performance of the GRACE metric to the choice of this gradient norm scaling? Do other scalings significantly affect the correlation/best teacher selection?
1. G-vendi seems to perform very well and similar to the author's method (Figure 13), except for the Qwen family. Do the authors have any conjecture as to why that may be?
1. Line 878 says "we use d=1024". Are these the results in Figure 8?
1. Does "GradCV" in Figure22/23 refer to GRACE?
1. The proposed method will penalize gradient variance - eg if a teacher were to generate "diverse" generations in some sense. In multi-task settings, or mixture of datasets (as is common in llm finetuning), will this penalty be counterproductive in selecting the right teacher, selecting teachers with less "coverage"?
1. (minor) Legend colors are missing in Figure 13.

---

> ### Author Response · Authors · 2025-11-23
>
> Thank you for the comments and suggestions about our paper. Please find responses to your questions below.
>
> **“Will GRACE penalize gradient variance and select teachers with less “coverage”?”**
>
> GRACE does not discourage diversity or teacher’s coverage; rather, it quantifies “coverage” using the span of student gradients where a better coverage is preferred. Specifically, GRACE balances gradient variance with the spectrum of the normalized gradient matrix. While the former may penalize excessive diversity, as the reviewer noted, the latter explicitly encourages diversity in the teacher-generated data. This balance is crucial because it ensures teacher-induced diversity is distributed across the training set rather than concentrated on a few prompts.
>
> Intuitively, GRACE assesses whether a teacher provides sufficiently varied responses such that, when a few samples are randomly removed, the remaining data preserves a well-spread spectrum. This is the key idea linking GRACE to the theory of conditional mutual information (Lemma 2.1). Thus, GRACE does not simply penalize variance—it evaluates how effectively the dataset’s overall coverage offsets it.
>
> **“Most of the experiments in the paper use temperature 1. However, the performance correlation of GRACE drops sharply for greedy sampling. Does that weaken the impact of experiments in the paper?”**
>
> We choose average-at-16 as the performance measure over greedy sampling because average-at-16 is a more stringent evaluation metric that tests consistency of generation from the model, as it tests the model’s ability to *consistently* solve a problem across multiple attempts. It has recently been used for assessing robust performance on Olympiad-level benchmarks and also serves as a key loss component when training models via reinforcement learning (e.g. DeepSeek-R1 [1]).
>
> Below, we also report how the performance of GRACE changes when we vary the temperature of evaluation from 0.0 to 1.0. Although the Spearman correlation shows a drop as temperature decreases, it remains higher than 0.7 across the temperatures. On the other hand, Performance Gap (Section C.2), which measures the difference between the best performing student and the student under the teacher selected by a score, remains roughly similar across all the temperatures. This shows that GRACE is robust to the temperature choice.
>
> | Temperature | **Absolute Spearman (↑)** | **Performance Gap (↓)** |
> | --- | --- | --- |
> | 0.0 | 0.7 | 2.4 |
> | 0.3 | 0.75 | 0.2 |
> | 0.6 | 0.83 | 1.2 |
> | 0.8 | 0.87 | 0.3 |
> | 1.0 | 0.86 | 0.3 |
>
> Comparisons with G-Var and G-Vendi.
> | **Temp** | **G-Var** (**Absolute Spearman (↑)**) | **G-Vendi** (**Absolute Spearman (↑)**) | **GRACE** (**Absolute Spearman (↑)**) | **G-Var** (PerformanceGap ↓) | **G-Vendi** (PerformanceGap ↓) | **GRACE** (PerformanceGap ↓) |
> | --- | --- | --- | --- | --- | --- | --- |
> | **0.0** | 0.34 | 0.63 | **0.70** | 13.5 | 4.5 | **2.4** |
> | **0.3** | 0.42 | 0.65 | **0.75** | 9.2 | 4.1 | **0.2** |
> | **0.6** | 0.51 | 0.62 | **0.83** | 9.5 | 8.5 | **1.2** |
> | **0.8** | 0.58 | 0.57 | **0.87** | 8.0 | 8.3 | **0.3** |
> | **1.0** | 0.44 | 0.56 | **0.86** | 10.8 | 14.5 | **0.3** |
>
>
> [1]: DeepSeek-R1: Incentivizing Reasoning Capability in LLMs via Reinforcement Learning. Guo et al. 2025
>
> **“How sensitive is the performance of the GRACE metric to the choice of the gradient norm scaling? Do other scalings significantly affect the correlation/best teacher selection?”**
>
> We find that scaling of gradients with length has a small effect on the results of GRACE. The results when we vary the scaling of gradients with length is given below:
>
> | Setting (Student model + dataset) | Without Length Scaling | With $\log T$ Length Scaling (i.e. GRACE) | With $\sqrt T$ Length Scaling | With $T$ Length Scaling |
> | --- | --- | --- | --- | --- |
> | LLaMA + GSM8K | 0.83 | 0.86 | 0.87 | 0.30 |
> | OLMo + GSM8K | 0.72 | 0.74 | 0.79 | 0.43 |
> | LLaMA + MATH | 0.86 | 0.89 | 0.78 | 0.06 |
> | Avg. | 0.80 | **0.83** | 0.81 | 0.26 |
>
> We find $\log T$ scaling to be the best overall, with no scaling or $\sqrt T$ scaling performing similarly. As the reviewer suggested, the scaling does appear to be dataset-dependent, and using dataset-specific scaling may further improve GRACE. However, to maintain generality and avoid overfitting to individual datasets, we use a single length-scaling choice across all datasets.
>
>
> **“How does the gradient norm distribution vary across datasets?”**
>
> In Figure 21, we presented the gradient norms for LLaMA-1B model on GSM8k dataset. In Figure 22 in the revision, we also present the gradient norms for LLaMA-3B model on MATH dataset and find that the gradient norms scale with $(\log T)^{2}$ slightly better. When we scale gradients with $(\log T)^{2}$, we observe the Spearman correlation of GRACE drops slightly to 0.84.

---

> ### Author Response · Authors · 2025-11-23
>
> **"Why does G-Vendi [Jung et al. 25] perform poorly on Qwen?"**
>
> The discrepancy is primarily driven by Qwen-1.5B-Instruct, which behaves as an outlier. When we exclude this teacher, the correlations in Figure 2 change as follows: G-Vendi improves from 0.44 to 0.58 (see Figure 24 in the revision), though it is still much worse than GRACE with a correlation of 0.86.
>
> Figure 13 reports *performance gap*: the difference between (i) the best performing student and (ii) the student trained using the teacher selected by a given score. As shown in Figure 25, once Qwen-1.5B-Instruct is removed, G-Vendi’s performance gap within the Qwen family improves from 14.5 to 1%. Importantly, however, this **excluded model is one that GRACE successfully identifies as poor.** We do not have a definitive explanation for why Qwen-1.5B behaves poorly: its response lengths and gradient norms are comparable to the other Qwen teachers.
>
>
>
> **We also appreciate the suggestion on the paper’s layout. We have updated the appendix accordingly in the revision. Some quick clarifications:**
>
> - “GradCV” (short for ”gradient cross-validation”) in Fig 22/23 is indeed an older name for GRACE. Thanks for catching this!
> - Sec C: which figures is the intro referring to? Where’s the GSM8k results?
>
> These are figures 9 and 10. We have modified accordingly.
>
> We have added an appendix overview and table of content at the top of the appendix. For now, we haven’t completely changed the layout of appendix to avoid confusion during the rebuttal phase but will modify in the final camera-ready version.

---

> > ### Comment · Reviewer_vXv9 · 2025-11-24
> > **Rebuttal Response Acknowledgement**
> >
> > **Regarding Sampling Temperature**
> >
> > Average-at-16 is definitely more robust than greedy sampling, and has indeed been used in DeepSeek-R1 etc. My concern was regarding the temperature (eg Deepseek R1 uses 0.6 with top-p 0.95, Section 3 of DeepSeek-R1 paper). That being said, the authors provide temperature ablation in the rebuttal which address my concerns, showing that their method is robust to decoding params.
> >
> > **Regarding Length Scaling**
> >
> > The authors show that using no-scaling or sqrt(T) length scaling with GRACE results in similar performance, hence addressing my concerns regarding the choice of scaling hyperparameter. **Was the same length-based gradient scaling also used for the baselines, eg G-Vendi? If yes, how do these perform without this scaling?**
> >
> >
> > **Regarding Qwen outlier for G-Vendi**
> >
> > Thank you for providing Figure 24 and 25, it address my concerns somewhat. Figure 25 does indeed highlight that G-vendi performs very well and somewhat similar to the author's method if the outlier of Qwen is excluded, but still achieves much worse correlation than GRACE.
> >
> > **ARC dataset results (in rebuttal response of Reviewer EhFJ)**
> >
> > These results are also quiet strong, highlighting the generalizability of this method. **Do the authors have any comment/conjectures as to why Teacher Performance performance so well on this dataset, but not as well on GSM8K/Math?**
> >
> > Thank you for the rebuttal, I have increased my score.

---

> > > ### Author Response · Authors · 2025-11-26
> > >
> > > We’re glad that our response has addressed your concerns, and thank you for engaging in the discussion!
> > >
> > > **Was the same length-based gradient scaling also used for the baselines, eg G-Vendi? If yes, how do these perform without this scaling?**
> > >
> > > Since G-Vendi uses normalized gradients, the length scaling doesn’t affect the result of G-Vendi.
> > >
> > > **“Do the authors have any comment/conjectures as to why Teacher Performance performance so well on ARC dataset, but not as well on GSM8K/Math?”**
> > >
> > > ARC dataset measures general reasoning.  ARC differs from GSM8K/Math by containing a large fraction of questions based on factual knowledges. For example, there are questions of the form
> > >
> > >     “A fold observed in layers of sedimentary rock most likely resulted from the:”
> > >
> > >     (a) cooling of flowing magma.
> > >     (b) converging of crustal plates.
> > >     (c) deposition of river sediments.
> > >     (d) solution of carbonate minerals.
> > >
> > > In such cases, a stronger teacher can directly convey the correct factual information, so Teacher Performance naturally correlates better with student performance. In contrast, GSM8K and MATH emphasize multi-step mathematical reasoning, where raw teacher accuracy is less informative; what matters instead is how compatible the teacher’s reasoning chains are for supervising the student.
> > >
> > > Thus, different metrics may work best depending on the task, but **GRACE remains consistently strong across datasets and student models.**

---

> > > > ### Comment · Reviewer_vXv9 · 2025-11-26
> > > > **Thank you for the clarifications**
> > > >
> > > > Thanks for the response, regarding normalization for GVendi, and the conjecture regarding teacher performance on ARC seems reasonable.

---

### Official Review · Reviewer_miyx · 2025-11-01

**Soundness:** 2
**Presentation:** 2
**Contribution:** 2
**Rating:** 4
**Confidence:** 3

**Summary:**

This paper addresses the problem of selecting the optimal teacher model for knowledge distillation, particularly in the context of large language models. The authors propose a score called GRACE that predicts a teacher's effectiveness for a given student and task. The score is computed using student gradients on a small set of teacher-generated data, without requiring access to teacher internals or test data. The authors provide a theoretical motivation for GRACE by connecting it to leave-one-out conditional mutual information and demonstrate empirically that it correlates strongly with final student performance.

**Strengths:**

1. The paper is well-motivated. The trial-and-error approach for teacher selection is computationally costly, and a principled, efficient method is a valuable contribution.

2. The connection of the proposed score to CMI for generalization bound is clever.

3. A key strength is that GRACE is lightweight and does not require access to teacher logits or internal states. This makes the method broadly applicable, even in black-box or API-based distillation scenarios.

**Weaknesses:**

1. The theoretical motivation (Lemma 2.1 and Appendix A) connects GRACE to the generalization gap in terms of loss. However, the empirical experiments measure correlation with final test accuracy (Average-at-k). The authors also note in the appendix (lines 752-755) that "GRACE serves as a reliable predictor of student performance, even though it fails to correlate with loss-based quantities". This is a disconnect that undermines the theoretical argument and should be discussed more prominently.

2. The paper's writing could improve. Particularly, Lemma 2.1 does not explain what are the variables in the conditional MI and which variable is the condition. This lemma needs to become well-defined.

3. There are a few claims in the paper that either I misunderstood or they are wrong. For example, lines 190-193 connect the small eigenvalues to high-signal direction. This is quite contrary to what eigenvalues express. Another example, In Fig 6, final sentence of the caption is not correct, it does not match the figure.

**Questions:**

1. In Fig 6, could you please clarify what the right figure is showing and how it should be compared to the other two figures? It seems that the results of the proposed metric does not match the right figure.

2. The authors nicely study the effect of their score on the correctness of the answer. How do you expect GRACE to perform on tasks where generation diversity is more important than correctness, such as dialogue or story generation?

3. Is there a principled way of choosing $C$ in your method? It seems that it is costly to run the trail and error.

4. The paper finds that filtering teacher generations for correctness is not beneficial for student performance (Fig 15). This is an interesting result. Do you have a hypothesis for why this happens? Could this perhaps suggest that the style or reasoning process is more important to distill than the final correct answer?

---

> ### Author Response · Authors · 2025-11-23
>
> Thank you for the comments and suggestions on the paper. We’d like to start with clarifications around GRACE.
>
>  **“Discussion on the gap between theoretical argument that uses CMI to define generalization gap in terms of loss and the application of GRACE to predict student test performance”**
>
> Our theoretical connection between GRACE and CMI is intended to formalize the core intuition behind the metric: that assessing gradient variance relative to the spectrum of the normalized gradient matrix provides a principled measure of how sensitive gradient updates are to the removal of individual samples, and therefore of a teacher’s utility.
>
> As the reviewer correctly points out, existing CMI bounds are designed around loss-based quantities such as cross-entropy, while our empirical evaluations center on downstream accuracy (Average-at-k). This mismatch reflects a broader limitation of current generalization theory rather than a weakness of GRACE. Developing CMI-style bounds that directly target downstream performance metrics will require new theoretical tools, and we leave this as an important direction for future work. We have clarified this difference both in the main paper (line 238 - 240) and in the appendix (line 806 - 808).
>
> **"Is there a principled way to Choose C for GRACE?"**
>
> We show in figure 20 in the appendix that any value C ≥ 6 essentially returns the same spearman correlation on both GSM8k and MATH dataset.
>
> **"Why does filtering for correctness not make a difference?"**
>
> This has also been shown in prior works that filtering for correctness doesn’t affect the student performance [1] or may even be counterproductive [2]. As hypothesized by the reviewer, these results also show that teaching the student model with diverse reasoning process is more important than teaching with correct responses. And this is what GRACE is also trying to measure — how diverse the gradients are across the teacher’s responses and how much variance there is as measured under the spectrum of the normalized gradients.
>
> [1]: Guha et al. 25: OpenThoughts: Data Recipes for Reasoning Models.
>
> [2]: Zhu et al. 25: The Surprising Effectiveness of Negative Reinforcement in LLM Reasoning
>
> **“How will GRACE perform on tasks where generation diversity is more important than correctness, such as dialogue or story generation?”**
>
> This is an interesting question! We did not evaluate with dialogue or stories since the evaluation is mostly qualitative and subjective (e.g. common evaluation are based on LLM-as-a-judge). Moreover, we’d like to clarify that semantic diversity does not necessarily translate to gradient diversity, as the latter might be more heavily affected by stylistic elements such as formatting and phrasing. Extending the scope of GRACE to include such tasks is an interesting future direction.
>
> **We also appreciate the suggestion on improving the writing, and we have updated accordingly in the revision.**
>
> To highlight a few clarifications:
>
> - Eigenvalues vs signal (line 190-193): Thanks for catching this typo! It should be that smaller eigenvalues are connected to higher-noise directions, as the reviewer pointed out. We’ve fixed this in the revision.
> - Lemma 2.1: The LOO-CMI is defined between the updated parameter and the left-out prompt, as specified on line 213. We have updated the lemma statement to clarify.
> - Figure 6: We have updated the figure to make the relation between temperature and performance clearer.

---

> > ### Comment · Reviewer_miyx · 2025-11-25
> > **Response to Authors' Rebuttal**
> >
> > Thank you for the detailed response and clarification. Most of my questions are now resolved. I do, however, have one remaining point of confusion after reading the authors’ rebuttal.
> >
> > I understand that the authors primarily use the theoretical result to motivate their approach and to provide insight into their proposed metric. However, in lines 752–755 of the previous draft (as noted in my main comment), the authors stated that “GRACE serves as a reliable predictor of student performance, even though it **fails** to correlate with loss-based quantities.” It appears that this sentence has been removed or rephrased in the current version. I am therefore wondering whether the authors could elaborate on this point, and clarify whether the metric does, in fact, correlate with loss-based quantities, as one might expect based on the theoretical results. If it does not, it would be helpful if the authors could provide some additional insight into why this might be the case.

---

> ### Author Response · Authors · 2025-11-26
>
> We thank the reviewer for engaging in the discussion!
>
> **Further clarification on “GRACE serves as a reliable predictor of student performance, even though it fails to correlate with loss-based quantities.”**
>
> We’d like to start with clarifying about the original wording: The pronoun “it” was intended to refer to the gap between loss-based quantities and *student performance*. We meant to say that **cross-entropy loss (both before and after training) does not correlate with student model’s performance, but GRACE correlates with student performance.** As correlations are transitive in nature, GRACE doesn’t correlate to loss-based quantities.
>
> We have revised the text in the new version to avoid this confusion.
>
> | Performance score | Absolute Spearman (↑) (to student performance) | Performance Gap (↓) |
> | --- | --- | --- |
> | Student Validation Loss before training | 0.07 | 5.6% |
> | Student Validation Loss after training | 0.15 | 4.1% |
> | Student Validation Loss - Train Loss after training | 0.17 | 4.1% |
> |  |  |  |
> |  |  |  |
>
> Table 1. In the GSM8K setting with LLaMA-1B, cross-entropy loss (pre- and post-training) on a teacher’s validation responses does not correlate with student performance, when compared across the teachers. Performance gap is given by the difference between (i) the best performing student and (ii) the student trained using the teacher selected by a given score.
>
> **Further details:**
>
> We outline two source of mismatch between the theoretical connection to CMI and the empirical results: the mismatch between the cross-entropy loss and accuracy, and the mismatch between Lemma 2.1 and GRACE’s weak correlation to the loss.
>
> For the mismatch between the student’s cross entropy (CE) loss and accuracy, there are two main reasons:
>
> - The CE losses are not directly comparable across teachers, whereas the accuracy is against a common ground truth. This is because the CE loss is computed against each teacher’s generations which can have distinct (chain-of-thought) styles, and the CE loss is sensitive to formats.
> - The CE loss measures token-level prediction error on a reference set of responses, whereas accuracy evaluation measures whether the model can *sequentially generate* a correct final answer, which is a global property.
>
> Thus, a student may achieve low cross-entropy loss on teacher-generated data after distillation yet still fail to produce correct solutions under evaluation. This discrepancy has been widely noted in prior work [1,2], which motivates the need for new CMI generalization bounds expressed in terms of *performance* rather than loss.
>
> For the second mismatch, i.e. between our theoretical statement and the empirical results, we note that GRACE is theoretically shown to upper bound CMI after a single gradient update, whereas empirically the loss is computed at the end of training which consists of multiple steps. Thus, as the reviewer pointed out, the framework of CMI is used as a motivation to understand GRACE, and we need stronger theoretical tools to fully characterize the success of GRACE.
>
> References:
>
> 1: What is wrong with perplexity for long-context language modeling? Fang et al.'24
>
> 2: Why exposure bias matters: An imitation learning perspective of error accumulation in language generation. Arora et al.'22

---

> > ### Comment · Reviewer_miyx · 2025-11-27
> > **Acknowledging Authors’ Rebuttal**
> >
> > Thank you for the clarification. I have raised my score.

---

### Official Review · Reviewer_EhFJ · 2025-11-01

**Soundness:** 3
**Presentation:** 3
**Contribution:** 2
**Rating:** 6
**Confidence:** 4

**Summary:**

The paper introduces GRACE, a lightweight score designed to optimize teacher selection for knowledge distillation in math problem-solving tasks.
GRACE quantifies the effectiveness of a teacher model by measuring the distributional properties of student gradients, without needing access to teacher logits, internals, or test data.
It is linked to generalization performance and correlates strongly with the performance of distilled student models (up to 86% on GSM8K and MATH datasets).
GRACE improves student performance by at least 6% over traditional methods that use the best-performing teacher.
Additionally, GRACE provides valuable insights on key distillation decisions, such as selecting the optimal teacher, generation temperature, and model size.
Overall, GRACE offers a more efficient and effective approach for teacher selection in distillation.

**Strengths:**

- GRACE offers an efficient method to identify the most suitable teacher for distillation, improving the distillation process without requiring extensive trial and error.

- The proposed GRACE strongly correlates with student performance, as demonstrated by its high correlation (up to 92%) with post-distillation results, out-performing traditional metrics like G-Vendi and G-Var.

- It provides actionable insights to practitioners, including optimal generation temperature and teacher selection under size or model family constraints.

**Weaknesses:**

- While effective, GRACE's correlations with student performance are not perfect, indicating that further refinement and additional explanatory factors might improve its accuracy.

- The score’s performance can be sensitive to hyperparameters, such as the number of prompts and generation dimension, which can complicate its application.

- The current task is limited to Math problem-solving, however, this is a small task within the NLP and AI domain. Is there any possibility of generalizing to other tasks or any insights benefiting other tasks?

- GRACE primarily relies on gradient information, which may not fully capture other important aspects of the distillation process, such as task-specific features. While not needing teacher logits is an advantage, in some cases, having access to them could potentially further improve the performance of GRACE. Would it be better to incorporate comprehensive information?

**Questions:**

- Is there any possibility of generalizing to other tasks or any insights benefiting other tasks?

- Would it be better to incorporate comprehensive information?

---

> ### Author Response · Authors · 2025-11-23
>
> **“How additional factors can be added to GRACE for perfect correlation?”**
>
> Our results show that GRACE can more reliably predict student’s post-distillation performance, compared against various methods based on gradients, and performance measures (e.g. student’s loss, teacher’s performance). Even though GRACE doesn’t achieve perfect correlation, GRACE can return a teacher that can train a student within 0.3% of the best performing student on GSM8k (figure 2). This suggests that correlation might not be the only metric to measure the utility of a given teacher score.
>
> In principle, correlation could be improved by incorporating additional terms that measure alignment between the teacher and the student, e.g. similarity between teacher and student intermediate representations, or alignment in their training distributions. However, this requires white-box access of the teacher or knowledge about the training datasets, which is more computationally expensive or not available (lines 480-484).
>
> **“Sensitivity to hyperparameter choice such as the number of prompts and generation dimension”**
>
> We find that choosing the gradient dimension to match the number of prompts yields the strongest performance across datasets. Section D.3 reports ablations showing how GRACE’s Spearman correlation varies with different choices of prompt count and gradient dimension. In our main experiments, we use 512 prompts and a 512-dimensional gradient that leads to the best correlation.
>
>
> **“Discussion on utility of GRACE to other domains beyond math reasoning”**
>
> We provide additional experiments on the ARC dataset [Clark et al. 2018] as attached below. GRACE achieves the best Spearman correlation among methods that do not require a verifier (i.e. all but Teacher performance), and achieves the lowest performance gap overall.
>
> | **Score** | **Absolute Spearman (↑)** | **Performance Gap (↓)** |
> | --- | --- | --- |
> | G-Vendi | 0.11 | 19.1% |
> | G-Var | 0.43 | 2.4% |
> | **GRACE** | **0.56** | **1.5%** |
> | Determinant | 0.09 | 28.9% |
> | BADGE | 0.19 | 28.9% |
> | Gradient inner product | 0.09 | 6.4% |
> | Gradient cosine | 0.19 | 28.9% |
> | Student’s loss on teacher’s generations | 0.14 | 12.9% |
> | Average (token) probabilities | 0.09 | 11.5% |
> | Average length | 0.06 | 6.4% |
> | Teacher performance | **0.82** | 2.4% |
>
> Here Performance Gap is given by the gap between the best student performance and the performance of the student under the teacher judged the best under a score (defined in appendix C.2). What we observe is that the teacher with lowest GRACE score will return a student within 1.5% of the best achievable performance. **Thus, while different scores may perform best on individual datasets, GRACE consistently delivers strong performance on average across all datasets and student models.**

---

### Author Response · Authors · 2025-12-03
**Rebuttal summary**

We thank the reviewers for their feedback and engagement during the rebuttal period, and summarize the discussions below.

Our paper proposes a score named **GRACE** for selecting teachers in LLM knowledge distillation. All reviewers recognize the **efficiency** and **effectiveness** of GRACE: GRACE strongly correlates with the student’s post-distillation performance, and identifies teachers that have a low performance gap to the post-hoc best teacher. GRACE offers “**actionable insights to practitioners**” (Reviewer EhFj): it can be used to guide distillation designs, such as choosing temperature, models within a family, or models under a size constraint.

Additionally, GRACE is noted to be “well motivated” (Reviewer miyx, vXv9) and “principled” (Reviewer “miyx”).

The discussions are centered on the following aspects:

- **Clarification on GRACE’s design choices**:
    - *How GRACE incorporates both coverage and diversity* (Reviewer vXv9): GRACE balances gradient variance with the spectrum of the normalized gradient matrix, in a cross-validation fashion.
    - *Connection to CMI* (Reviewer miyx): GRACE’s cross-validation design is similar to that of conditional mutual information (CMI), which captures a core intuition of GRACE that it provides a principled measure of how sensitive gradient updates are to the removal of individual samples, and therefore of a teacher’s utility.
    - *Towards perfect correlation* (Reviewer EhFJ): While we agree that including other factors (e.g. using verifiers, or knowledge of teacher’s internals or training details) can be informative, GRACE intentionally avoids these information to be more generally applicable. Also, we find that using verifiers to filter for correct generations (Reviewer miyx) is not helpful empirically, which is likely because incorrect generations can still be partially correct and provide useful diversity.
- **Robustness of the empirical results**: this includes the generalization to non-math datasets (Reviewer EhFJ), and sensitivity to hyperparameters (Reviewer vXv9). We provided additional results in the rebuttal which have convinced the reviewers that GRACE is robust:
    - We provided additional results on ARC (which focuses on general reasoning) in [the response to Reviewer EhFJ](https://openreview.net/forum?id=m276fke38H&noteId=vy9beGxd83): GRACE achieves the best Spearman correlation among methods that do not require a verifier (i.e. all but Teacher performance), and achieves the lowest performance gap overall.
    - We provided ablation results with various generation temperatures and gradient length normalization in [response to Reviewer vXv9](https://openreview.net/forum?id=m276fke38H&noteId=DO1oUe4LF7), where GRACE outperforms the baselines consistently.
- **Presentation and writing clarity**: We appreciate the suggestions and have updated the paper accordingly. Changes are highlighted in blue.

We believe the reviewers’ concerns have been addressed in the rebuttals and all reviewers support to accept the paper.

---

### Meta-Review · Area_Chair_uwNT · 2026-01-06

**Summary:**

This paper proposes GRACE, a gradient-based score for teacher selection in knowledge distillation. The method is lightweight and does not require teacher internals or test data. Reviewers generally agree that the proposed method is well motivated and empirically effective. The reviewers mainly raised two points: the gap between the CMI-based theoretical motivation and the accuracy-based empirical evaluation, and whether the method can generalize beyond math reasoning tasks. The authors addressed these points in the rebuttal and added new experiments, which largely clarified the concerns. Overall, these issues no longer outweigh the demonstrated empirical effectiveness, and the paper is recommended for acceptance.

**Reviewer Concerns:**

The rebuttal addressed many technical and clarity-related issues. The authors clarified theoretical motivations and fixed presentation errors. Additional ablations and ARC experiments improved confidence in robustness. However, some concerns remain partially unresolved. The theoretical gap between CMI-based arguments and accuracy metrics persists. Evidence for generalization beyond reasoning-style tasks remains limited.

**Reviewer Scores:**

Reviewer EhFJ was generally positive, with concerns about robustness and generalization. These concerns appear to have been largely addressed in the rebuttal, and the score is likely unchanged or slightly improved. Reviewer miyx initially raised concerns about theoretical clarity and presentation, which were mostly resolved in the rebuttal, leading to an increased score. Reviewer vXv9 questioned the experimental choices and robustness, and found the additional analyses provided in the rebuttal convincing, also resulting in a higher score. Overall, the scores are likely to remain mixed but show a more positive trend.

---

### Decision · Program_Chairs · 2026-01-26

Accept (Poster)